# GRAND++: Graph Neural Diffusion with A Source Term

**Matthew Thorpe\*[1], Hedi Xia\*[2], Tan Nguyen\*[2], Thomas Strohmer[3]**
**Andrea L. Bertozzi[2], Stanley J. Osher[2] & Bao Wang[4]** *
[1]Department of Mathematics, University of Manchester, Manchester M13 9PL, UK
[2]Department of Mathematics, UCLA, Los Angeles, CA, 90095, USA
[3]Department of Mathematics, UC Davis, Davis, CA 95616, USA
[4]Department of Mathematics and Scientific Computing and Imaging (SCI) Institute
University of Utah, Salt Lake City, UT, 84102, USA

## Abstract

We propose GRAph Neural Diffusion with a source term (GRAND++) for graph deep learning with a limited number of labeled nodes, i.e., low-labeling rate. GRAND++ is a class of continuous-depth graph deep learning architectures whose theoretical underpinning is the diffusion process on graphs with a source term. The source term guarantees two interesting theoretical properties of GRAND++: (i) the representation of graph nodes, under the dynamics of GRAND++, will not converge to a constant vector over all nodes even as the time goes to infinity, which mitigates the over-smoothing issue of graph neural networks and enables graph learning in very deep architectures. (ii) GRAND++ can provide accurate classification even when the model is trained with a very limited number of labeled training data. We experimentally verify the above two advantages on various graph deep learning benchmark tasks, showing a significant improvement over many existing graph neural networks.

## 1 Introduction

Graph neural networks (GNNs) are the backbone for deep learning on graphs. Recent GNN architectures include graph convolutional networks (GCNs) [30], ChebyNet [16], GraphSAGE [29], neural graph fingerprints [20], message passing neural networks [28], and graph attention networks (GATs) [54]. These graph deep networks have achieved success in many applications, including computational physics and computational chemistry [20, 28, 3], recommender systems [41, 62], and social networks [63, 47]. Hyperbolic GNNs have also been proposed to enable certain kinds of data embedding with much smaller distortion [11, 37]. See [6] for some recent advances of GNN algorithm development and applications.

A well-known problem of GNNs is that increasing the depth of GNNs often results in a significant drop in performance on various graph learning tasks. This performance degradation has been widely interpreted as the *over-smoothing* issue of GNNs [35, 44, 12]. Intuitively, GNN layers update the node representation by taking a weighted average of its neighbors' features, making representations for neighboring nodes to be similar. As the GNN architecture gets deeper, all nodes' representation will become indistinguishable resulting in over-smoothing. In Sec. 2, we briefly show that certain GNNs have a diffusive nature which makes over-smoothing inevitable. Another interesting interpretation of the GNN performance degradation is via a *bottleneck* [1], since a GNN tends to represent exponentially growing information from neighbors with fixed-size vectors. Several algorithms have been proposed to mitigate the over-smoothing of GNNs, including skip connection and dilated convolution [33], Jumping Knowledge [60], DropEdge [49], PairNorm [64], graph neural diffusion (GRAND) [10], and wave equation motivated GNNs [21]. Nevertheless, developing deep GNN architectures is still in its infancy compared to the development of other deep networks.

Besides suffering from over-smoothing, we notice that the accuracy of existing GNNs drops severely when they are trained with a limited labeled data. As illustrated in Fig. 1, the test accuracy of several

---

*Correspond to `wangbaonj@gmail.com` or `matthew.thorpe-2@manchester.ac.uk`

celebrated GNN architectures, including GCN, GAT, and GraphSage, drops rapidly when they are trained with fewer labeled data. Moreover, the variance of classification accuracy grows significantly as number of labeled nodes drops. Indeed, semi-supervised graph learning with very low-labeling rates has been studied in the Laplace learning and graph deep learning settings, see, e.g., [36, 9, 23]; one question is *can we develop new GNN architectures to improve the performance of graph deep learning in low-labeling rate regimes?*

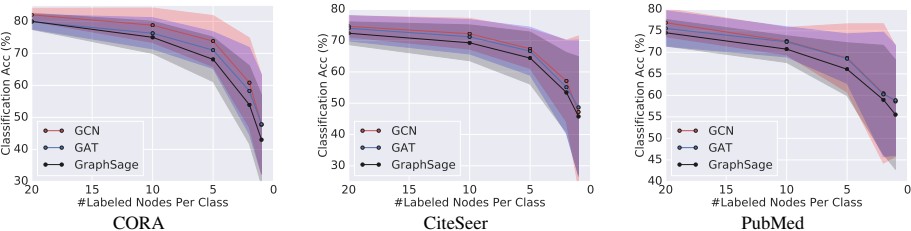

Figure 1: Test accuracy of GCN, GAT, and GraphSage vs. the number of labeled nodes per class. All networks have 2 layers, and each experiment is run with 100 splits and 20 random seeds following [10]. The accuracy drops rapidly with fewer labeled data for training. CORA, CiteSeer, and PubMed have 2485, 2120, and 19717 nodes in total respectively. Results on more benchmark GNN architectures are in Appendix D.4.

## 1.1 OUR CONTRIBUTION

With the above GNN problems in mind, we focus on developing new continuous-depth GNNs to overcome over-smoothing and boost the accuracy of GNNs with a limited number of labeled data. We first present a random walk interpretation of the GRAND model [10], revealing a potentially inevitable over-smoothing phenomenon when GRAND is implicitly very deep. Based on the random walk viewpoint of GRAND, we then propose graph neural diffusion with a source term (GRAND++) that corrects the bias arising from the diffusion process, see Sec. 5 for details. GRAND++ theoretically guarantees that: (i) under GRAND++ dynamics, the graph node features do not converge to a constant vector over all nodes even as the time goes to infinity, and (ii) GRAND++ can provide accurate prediction even when it is trained with a limited number of labeled nodes. Moreover, these theoretical results resonate with the practical advantages of GRAND++. We summarize the major practical advantages of GRAND++ below.

- GRAND++ can effectively overcome the over-smoothing issue; it is remarkably more accurate than existing GNNs when the architecture is very deep.
- GRAND++ is suitable for graph deep learning when only a few nodes are labeled as training data. Moreover, in the low-labeling rates, GRAND++ can be more accurate when the network is deeper.
- GRAND++ inherits the continuous-depth merit from GRAND, which defines the network depth implicitly and enables memory-efficient training by using the adjoint method.

## 1.2 RELATED WORK

**Diffusion on graphs and continuous-depth graph neural networks.** Diffusion has been defined on graphs, see, e.g., [25, 24], and used in various applications, including data clustering and dimension reduction [15, 4], image processing [27, 22, 17, 38], and semi-supervised graph nodes classification [67, 65]. From the numerical viewpoint, fast algorithms have been proposed for using diffusion on graphs to solve penalized graph cut problems [26]. The connection between GNNs and diffusion on graphs has been studied substantially. For instance, GNN has been interpreted as a diffusion process on graphs, which performs low-pass filtering on the input features [43]. Moreover, insights from the diffusion process on graphs have been used to improve the performance of GNNs, see, e.g., [2, 36, 31, 57].

Leveraging neural ordinary differential equations (ODEs) [13], continuous-depth GNNs have been proposed, see, e.g., [45, 58, 68]. One recent work is GRAND [10], which parameterizes the diffusion equation on graphs with a neural network. See Sec. 3 for a brief review of GRAND.

**Neural ODEs.** Neural ODEs [13] are a class of continuous-depth neural networks whose depth is defined implicitly. Training neural ODEs using the adjoint method [46] is more memory efficient than training other neural networks using backpropagation. We provide a brief review of neural ODEs and the adjoint method in Appendix C. GRANDs [10] are a class of neural partial differential equations (PDEs) on graphs that can also be considered as a coupled system of neural ODEs. Furthermore, GRANDs are also trained by using the adjoint method.

**Laplace learning and Poisson learning.** Laplace learning has been used for semi-supervised data classification [67, 65, 56], image processing [7, 27], etc. Direct application of Laplace learning with Gaussian weights [5] or locally linear embedding weights [50] for the above tasks may cause inference inconsistency when only a limited number of graph nodes are labeled, resulting in poor performance. Several algorithms address the inference inconsistency at low labeling rate. They include up-weighting the weights of the labeled data [52] and the $p$-Laplacian [8, 48, 66]. In [9], the authors have proposed Poisson learning for improving Laplace learning at extremely low-labeling rate regimes. Poisson learning augments Laplace learning with a Green's function at each labeled data, enabling accurate node classification when only a few labeled data are available. Compared to Laplace learning, Poisson learning adds Green's function to the label of each labeled node and then performs label propagation to predict the label for unlabeled graph nodes. GRAND and GRAND++ both learn graph node representations and perform prediction by activating the node representations, which are fundamentally different from Laplace and Poisson learning.

### 1.3 NOTATION

We denote scalars by lower- or upper-case letters and vectors and matrices by lower- and upper-case boldface letters, respectively. For a matrix $A$, we denote its transpose as $A^\top$ and its Hadamard product with another matrix $B$ as $A \odot B$, i.e., the entrywise multiplication of $A$ and $B$. We write the set $\{1, 2, \cdots, n\}$ as $[n]$. We denote the probability and expectation of a given random variable $x$ as $\mathbb{P}(x)$ and $\mathbb{E}[x]$, respectively. The meaning of other notations can be inferred from the context.

### 1.4 ORGANIZATION

The paper is organized as follows: In Sec. 2, we review diffusion equation on graphs and its connection to GNNs. In Secs. 3 and 4, we briefly review GRAND and present a random walk interpretation of GRAND, respectively. Leveraging the random walk viewpoint of GRAND, we propose GRAND++ for deep graph learning with theoretical guarantees in Sec. 5. We verify the efficacy of GRAND++ in Sec. 6. Technical proofs and more validations are provided in the appendix.

## 2 BACKGROUND

**Diffusion equation on graphs.** Let $G = (X, W)$ represent an undirected graph with $n$ nodes, where $X = \left([x^{(1)}]^\top, \cdots, [x^{(n)}]^\top\right)^\top \in \mathbb{R}^{n \times d}$ with each row $x^{(i)} \in \mathbb{R}^d$ a feature vector and $W := \left(W_{ij}\right)$ a $n \times n$ matrix with $W_{ij}$ representing the similarity (edge weight) between the $i^{th}$ and $j^{th}$ feature vectors, and we assume $W_{ij} = W_{ji}$. Consider the following diffusion process that evolves the feature matrix $X$ on the graph (see Appendix A for a brief review of calculus on graphs):

$$\frac{\partial X(t)}{\partial t} = \text{div}\big(G(X(t), t) \odot \nabla X(t)\big), \tag{1}$$

where $X(t) = \left([x^{(1)}(t)]^\top, \cdots, [x^{(n)}(t)]^\top\right)^\top \in \mathbb{R}^{n \times d}$ with $x^{(i)}(0) = x^{(i)}$, $\nabla$ and div are the gradient and divergence operators, respectively. The matrix $G(X(t), t)$ is chosen such that $W \odot G$ is right-stochastic, i.e., each row of $W \odot G$ summing to 1. In the machine learning setting, we can parameterize $G$ with learnable parameters $\theta$ which we denote by $G(X(t), t, \theta)$. The initial features are evolved under the diffusion dynamics (1) from $t = 0$ to $T$ to learn the final representation $X(T)$ for further machine learning tasks.

In the simplest case when $G(X(t), t)$ is only dependent on the initial node features $X$, i.e., $G$ is time-independent, right-stochasticity implies $\sum_j W_{ij}G_{ij} = 1$ for all $i$, and so we focus on the particular case when $G_{ij} = 1/d_i$ with $d_i = \sum_{j=1}^n W_{ij}$. In this case the right-hand side of (1) reduces to the negative of the random-walk Laplacian applied to $X(t)$ and (1) becomes

$$\frac{\partial X(t)}{\partial t} = \text{div}\big(G(X(t), t) \odot \nabla X(t)\big) = -LX(t), \tag{2}$$

where $L = I - D^{-1}W := I - A\,(A := A(X))$ is the random walk Laplacian and $D$ is diagonal with $D_{ii} = d_i$. See [14, 25, 24] for more about random walk Laplacian and diffusion on graphs.

**Graph neural networks.** Applying forward Euler discretization, with step size $\delta_t < 1$, of (2) gives

$$X(k\delta_t) = X((k-1)\delta_t) - \delta_t L X((k-1)\delta_t) := \tilde{L}X((k-1)\delta_t), \text{ for } k = 1, 2, \cdots, K, \tag{3}$$

$T = K\delta_t$, and $\boldsymbol{X}(0) = \boldsymbol{X}$. Note that the matrix $\tilde{\boldsymbol{L}}$ is the discretization of the diffusion operator, which is a special low-pass filter. Equation (3) is a prototype for motivating GNNs: by introducing weights $\boldsymbol{W}^{(k)} \in \mathbb{R}^{d \times d}$ and a nonlinearity $\sigma$, e.g., ReLU, into (3), we have

$$\boldsymbol{X}((k+1)\delta_t) = \sigma\big(\tilde{\boldsymbol{L}}\boldsymbol{X}(k\delta_t)\boldsymbol{W}^{(k)}\big). \tag{4}$$

The model in (4) is similar to the well-established GCN architecture proposed in [30]. The diffusive nature of the GNN architecture in (4) further explains the over-smoothing issue of training deep GNNs; the deeper the network architecture is, the more the node features diffuse. Eventually, all nodes share similar features and become indistinguishable. See Sec. 5 for a detailed analysis.

## 3 A BRIEF REVIEW OF GRAND

GRAND is a new continuous-depth GNN proposed in [10]. It integrates a learnable encoder function $\phi$ and a learnable decoder function $\psi$ with the neural network parameterized graph diffusion process, resulting in the prediction $\boldsymbol{Y} = \psi(\boldsymbol{X}(T))$, where $\boldsymbol{X}(T)$ is computed as

$$\boldsymbol{X}(T) = \boldsymbol{X}(0) + \int_0^T \frac{\partial \boldsymbol{X}(t)}{\partial t} dt, \text{ with } \boldsymbol{X}(0) = \phi(\boldsymbol{X}), \tag{5}$$

where $\partial \boldsymbol{X}(t)/\partial t$ is given by the graph diffusion equation (2). From the neural ODE perspective, we can perform forward propagation of GRAND, i.e., we solve (5), using numerical ODE solvers.

In the simplest case, when $\boldsymbol{G}$ is only dependent on the initial node features, we can rewrite (1) as

$$\frac{\partial \boldsymbol{X}(t)}{\partial t} = \big(\boldsymbol{A}(\boldsymbol{X}) - \boldsymbol{I}\big)\boldsymbol{X}(t), \tag{6}$$

GRAND models the diffusivity $\boldsymbol{A}(\boldsymbol{X})$ in (6) by the multi-head self-attention mechanism; potential choices of the attention function include the ones proposed in [53, 54]. More precisely, in GRAND $\boldsymbol{A}(\boldsymbol{X}) = \frac{1}{h}\sum_{l=1}^h \boldsymbol{A}^l(\boldsymbol{X})$ with $h$ being the number of heads and the attention matrix $\boldsymbol{A}^l(\boldsymbol{X}) = (a^l(\boldsymbol{x}_i, \boldsymbol{x}_j))$, for $l = 1, \cdots, h$, is computed as follows:

$$a^l(\boldsymbol{x}_i, \boldsymbol{x}_j) = \frac{\exp\big(\text{LeakyReLU}(\boldsymbol{a}^{l\top}[\boldsymbol{W}^l\boldsymbol{x}_i \| \boldsymbol{W}^l\boldsymbol{x}_j])\big)}{\sum_{k \in \mathcal{N}_i} \exp\big(\text{LeakyReLU}(\boldsymbol{a}^{l\top}[\boldsymbol{W}^l\boldsymbol{x}_i \| \boldsymbol{W}^l\boldsymbol{x}_k])\big)}, \tag{7}$$

where $\boldsymbol{W}^l$ and $\boldsymbol{a}^l$ are learned, $\|$ is the concatenation operator, and $\mathcal{N}_i$ is the index set of the nodes that are connected to the $i^{th}$ node in the graph. GRAND with the attention in (7) is called GRAND-l, that is, GRAND-l is a special case of GRAND when the diffusivity is dependent only on the initial graph node features. Time-dependent attention and graph rewiring can be integrated into GRAND, resulting in GRAND-nl and GRAND-nl-rw, respectively [10]. From the ODE viewpoint, GRAND and its variants are a class of coupled neural ODEs defined on an unweighted graph. Their merits include continuous-depth and memory-efficient training using the adjoint method [46, 13].

## 4 RANDOM WALK VIEWPOINT OF GRAND

In this section, we present a random walk interpretation of GRAND. The connection between graph random walks and the diffusion equation has been extensively studied, but we recap the key idea here to motivate the new GRAND with a source term architecture. Let $\{\boldsymbol{B}^{(i)}(k)\}_{k \in \mathbb{N}}$ be the random walk on $\{\boldsymbol{x}^{(j)}(0)\}_{j=1}^n$ defined by, for $\delta_t \in [0, 1]$,

$$\boldsymbol{B}^{(i)}(0) = \boldsymbol{x}^{(i)}(0)$$

$$\mathbb{P}\big(\boldsymbol{B}^{(i)}(k+1) = \boldsymbol{x}^{(\ell)}(0)|\boldsymbol{B}^{(i)}(k) = \boldsymbol{x}^{(j)}(0)\big) = \begin{cases} 1 - \delta_t & \text{if } \ell = j \\ \frac{\delta_t W_{j\ell}}{d_j} & \text{if } \ell \neq j \end{cases} \tag{8}$$

where $d_j = \sum_{\ell=1}^n W_{j\ell}$ (assume $W_{\ell\ell} = 0$ for all $\ell$). Proposition 1 below is well-known, see [67]. We provide the proof of Proposition 1 and all the subsequent theoretical results in Appendix B.

**Proposition 1** *Let $\boldsymbol{X}$ solve (3) and $\boldsymbol{B}^{(i)}$ be the random walk determined by (8) where $\delta_t \in [0, 1]$. Then*

$$\boldsymbol{x}^{(i)}(\delta_t k) = \mathbb{E}\big[\boldsymbol{B}^{(i)}(k)\big].$$

Proposition 2 below gives the stationary distribution of the random walk $\{\boldsymbol{B}^{(i)}(k)\}_{k\in\mathbb{N}}$.

**Proposition 2** *Assume the graph $G = (\boldsymbol{X}, \boldsymbol{W})$ is connected. Then, the stationary distribution of $\{\boldsymbol{B}^{(i)}(k)\}_{k\in\mathbb{N}}$ is*

$$\pi = \left( \frac{d_1}{\sum_{j=1}^{n} d_j}, \cdots, \frac{d_n}{\sum_{j=1}^{n} d_j} \right), \tag{9}$$

*which is independent of the starting position $\boldsymbol{x}^{(i)}$.*

Furthermore, we have the following theoretical result on the asymptotic behavior of graph node features under the GRAND dynamics given by (3).

**Proposition 3** *Assume the graph $G = (\boldsymbol{X}, \boldsymbol{W})$ is connected. Then for all $i = 1, \cdots, n$, we have*

$$\boldsymbol{x}^{(i)}(k\delta_t) \to \widetilde{\boldsymbol{x}} := \sum_{j=1}^{n} \boldsymbol{x}^{(j)}(0)\pi_j, \quad as \quad k \to \infty.$$

Hence, for the case of (3), i.e., GRAND-l, we expect the output to be approximately independent of the input, due to over-smoothing. Of course, once we reintroduce the $\boldsymbol{X}(t)$ dependence back into $\boldsymbol{G}$ in (1) and (2) or into the operator $\boldsymbol{A}$ in (6) then the above arguments no longer hold. Nevertheless, the GRAND architectures are built on a principle that is ill-suited to deep networks. In the next section we introduce a source term and perform a similar random walk analysis that illustrates how the new architecture can be better suited for deep GNN architectures.

## 5 GRAND++: Graph Neural Diffusion with A Source Term

### 5.1 Algorithm and formulation

At the core of GRAND++ is the introduction of a source term into GRAND, leveraging the random walk viewpoint of the diffusion process. We take a small subset of feature vectors, indexed by $\mathcal{I} \subseteq [n]$, believed to be "trustworthy" for use as a source term. In particular, we use the features of labeled data. The GRAND++ dynamics are defined by a diffusion equation with a source term (we use the variable $\boldsymbol{z}$ for GRAND++-related dynamics and $\boldsymbol{x}$ for GRAND dynamics)

$$\frac{\partial \boldsymbol{z}^{(i)}(t)}{\partial t} = \text{div}\left[\boldsymbol{G}(\boldsymbol{Z}(t), t) \odot \nabla \boldsymbol{Z}(t)\right]^{(i)} + \sum_{j\in\mathcal{I}} \delta_{ij} C_j \tag{10}$$

where $C_j$ is the source at feature vector of node $j$. Below we motivate a particular choice of $C_j$.

The key idea is to first characterise the bias that arises from the diffusion and use that to propose a correction via the choice of source terms $C_j$. Following the simplifications in (2), our diffusion equation (without the source term) follows the approximate dynamics when $t \gg 1$

$$\frac{\partial \boldsymbol{x}^{(i)}(t)}{\partial t} = -[\boldsymbol{L}\boldsymbol{X}(t)]^{(i)} = -\underbrace{\boldsymbol{x}^{(i)}(t)}_{\approx \widetilde{\boldsymbol{x}}} + \frac{1}{d_i}\sum_{j=1}^{n} W_{ij} \underbrace{\boldsymbol{x}^{(j)}(t)}_{\approx \widetilde{\boldsymbol{x}}} \approx 0.$$

For $i \in \mathcal{I}$, it transpires that choosing $C_i = \boldsymbol{x}^{(i)} - \hat{\boldsymbol{x}}$ (where $\hat{\boldsymbol{x}}$ is defined below) gives rise to a random walk interpretation that allows us to prove that the oversmoothing seen in the GRAND model is avoided.

One can in fact choose $\widetilde{\boldsymbol{x}}$ with a certain degree of freedom. If we initialise $\boldsymbol{X}(0) = \boldsymbol{X}$ then we obtain $\widetilde{\boldsymbol{x}} = \sum_{j=1}^{n} \boldsymbol{x}^{(j)}\pi_j$ (as is usual in the GRAND model). However, as the similarities are encoded in the graph weights, and the diffusion dynamics will drive it towards a non-trivial state, we can choose a different initialization than $\boldsymbol{X}(0) = \boldsymbol{X}$. Through connections with random walks we, in the next subsection, motivate an alternative initialisation

$$\sum_{i=1}^{n} \boldsymbol{z}^{(i)}(0) = \sum_{i\in\mathcal{I}} \frac{\boldsymbol{x}^{(i)} - \hat{\boldsymbol{x}}}{d_i}, \quad \text{where } \hat{\boldsymbol{x}} = \frac{1}{|\mathcal{I}|} \sum_{j\in\mathcal{I}} \boldsymbol{x}^{(j)} \tag{11}$$

with the dynamics

$$\frac{\partial \boldsymbol{z}^{(i)}(t)}{\partial t} = \text{div}\left[\boldsymbol{G}(\boldsymbol{Z}(t), t) \odot \nabla \boldsymbol{Z}(t)\right]^{(i)} + \sum_{j \in \mathcal{I}} \delta_{ij} \left(\boldsymbol{x}^{(i)} - \hat{\boldsymbol{x}}\right). \tag{12}$$

For example, we could choose

$$\boldsymbol{z}^{(i)}(0) = \begin{cases} \frac{1}{d_i}\left(\boldsymbol{x}^{(i)} - \hat{\boldsymbol{x}}\right) & \text{if } i \in \mathcal{I} \\ \boldsymbol{0} & \text{otherwise,} \end{cases} \quad \text{or} \quad \boldsymbol{z}^{(i)}(0) = \boldsymbol{x}^{(i)} - c,$$

where $c = \frac{1}{n}\left(\sum_{i=1}^n \boldsymbol{x}^{(i)} - \sum_{j \in \mathcal{I}} \frac{\boldsymbol{x}^{(j)} - \hat{\boldsymbol{x}}}{d_j}\right)$ is chosen such that (11) holds. We do not believe that the constant $c$ (that shifts by a constant) is particularly important but it is included to provide a random walk interpretation which helps to understand the deep architecture (when $T$ is big) behaviour of the model GRAND++. The justification for this choice will be made in Sec. 5.2. To summarize, the GRAND++ model in (12), with initial condition satisfying (11), simply adds a source term to the original GRAND model and uses a different initial condition. Therefore, the nonlinear diffusivity and graph rewiring tricks used by GRAND can be easily integrated into GRAND++. In terms of implementation, since GRAND++ merely changes the right-hand side of GRAND, which again can be regarded as a system of coupled first-order neural ODEs; we can leverage neural ODE training, testing, and inference for GRAND++ similar to GRAND.

In the next subsection we explore the random walk connection of the above model, suggesting that building a graph neural network based on the diffusion with source model does not suffer from the same degeneracy as we observed in Sec. 4 and is therefore better suited to build deep GNNs. In particular, we can write the diffusion with source model as the short time expected behaviour of a random walk and therefore we do not have the issue of reaching the stationary state (in other words passing the mixing time). Our experiments in Sec. 6 suggest the formal motivation holds and we are able to design deep GNNs.

## 5.2 THE RANDOM WALK PERSPECTIVE OF GRAND++

Let us continue to consider the simplified model in the previous subsection, i.e., assume the dynamics are governed by

$$\frac{\partial \boldsymbol{z}^{(i)}(t)}{\partial t} = -\left[\boldsymbol{L}\boldsymbol{Z}(t)\right]^{(i)} + \sum_{j \in \mathcal{I}} \delta_{ij} \left(\boldsymbol{x}^{(i)} - \hat{\boldsymbol{x}}\right) \tag{13}$$

where the initial condition satisfies (11). Using the forward Euler discretisation of the above dynamics we have

$$\boldsymbol{z}^{(i)}(\delta_t k) = \boldsymbol{z}^{(i)}(\delta_t(k-1)) - \delta_t \left[\boldsymbol{L}\boldsymbol{Z}(\delta_t(k-1))\right]^{(i)} + \delta_t \sum_{j \in \mathcal{I}} \delta_{ij} \left(\boldsymbol{x}^{(i)} - \hat{\boldsymbol{x}}\right), \tag{14}$$

for $k = 1, 2, \ldots, K$ where again $T = K\delta_t$.

We use the same random walk as that introduced in Sec. 4, i.e. the random walk defined by (8), but we will now only consider random walks that are initialised on the nodes indexed by $\mathcal{I}$.

**Proposition 4** *Let $\boldsymbol{Z}$ solve (14) with the initial condition satisfying (11), and let $\boldsymbol{B}^{(i)}$ be the random walk determined by (8). Then,*

$$\left|\boldsymbol{z}^{(i)}(k\delta_t) - \mathbb{E}\left[\sum_{s=0}^k \frac{1}{d_i} \sum_{j \in \mathcal{I}} \left(\boldsymbol{x}^{(j)} - \hat{\boldsymbol{x}}\right) \mathbb{1}_{\boldsymbol{B}^{(j)}(s) = \boldsymbol{x}^{(i)}}\right]\right| \to 0 \quad \text{as } k \to \infty.$$

**Remark 1** *In the limit $k \to \infty$ the term*

$$\mathbb{E}\left[\sum_{s=0}^k \frac{1}{d_i} \sum_{j \in \mathcal{I}} \boldsymbol{x}^{(j)} \mathbb{1}_{\boldsymbol{B}^{(j)}(s) = \boldsymbol{x}^{(i)}}\right]$$

*is formally a function of the random walk at all times. Whilst if $k$ is very large (i.e. in comparison to the mixing time) we still have that*

$$\mathbb{E}\left[\frac{1}{d_i} \sum_{j \in \mathcal{I}} \boldsymbol{x}^{(j)} \mathbb{1}_{\boldsymbol{B}^{(j)}(k) = \boldsymbol{x}^{(i)}}\right] = \frac{1}{d_i} \sum_{j \in \mathcal{I}} \boldsymbol{x}^{(j)} \underbrace{\mathbb{P}\left(\boldsymbol{B}^{(j)}(k) = \boldsymbol{x}^{(j)}\right)}_{\approx \pi_i} \approx \frac{\pi_i}{d_i} \sum_{j \in \mathcal{I}} \boldsymbol{x}^{(j)} \tag{15}$$

*and on the other hand*

$$\mathbb{E}\left[\frac{1}{d_i} \sum_{j \in \mathcal{I}} \hat{\boldsymbol{x}} \mathbb{1}_{\boldsymbol{B}^{(j)}(k) = \boldsymbol{x}^{(i)}}\right] = \frac{\hat{\boldsymbol{x}}}{d_i} \sum_{j \in \mathcal{I}} \underbrace{\mathbb{P}\left(\boldsymbol{B}^{(j)}(k) = \boldsymbol{x}^{(j)}\right)}_{\approx \pi_i} \approx \frac{\pi_i |\mathcal{I}| \hat{\boldsymbol{x}}}{d_i}. \tag{16}$$

*From the definition of $\hat{\boldsymbol{x}}$ we see that (15) and (16) are approximately equal. And therefore we can understand $\mathbb{E}[\frac{1}{d_i} \sum_{j \in \mathcal{I}} \hat{\boldsymbol{x}} \mathbb{1}_{\boldsymbol{B}^{(j)}(k) = \boldsymbol{x}^{(i)}}]$ as the long time behaviour of $\mathbb{E}[\frac{1}{d_i} \sum_{j \in \mathcal{I}} \boldsymbol{x}^{(j)} \mathbb{1}_{\boldsymbol{B}^{(j)}(k) = \boldsymbol{x}^{(i)}}]$ Very formally we can see that subtracting the long-time behaviour from the all-time behaviour leaves us with the short time behaviour. This provides one explanation as to why we do not expect the deep layers to be determined by the stationary state of the random walk (at which point there is little dependence on the initial layers, causing the deep layers to be approximately constant).*

**Remark 2** *The random walk interpretation*

$$\mathbb{E}\left[\sum_{s=0}^{k} \frac{1}{d_i} \sum_{j \in \mathcal{I}} \left(\boldsymbol{x}^{(j)} - \hat{\boldsymbol{x}}\right) \mathbb{1}_{\boldsymbol{B}^{(j)} = \boldsymbol{x}^{(i)}}\right] \tag{17}$$

*can be considered to be dual to the random walk interpretation in Sec. 4: in Sec. 4 we released the random walker from the node of interest, whilst now we release the random walkers from nodes indexed by $\mathcal{I}$ and see how many of them hit the node of interest. We note also that we do not require a lower bound on the size of the set $\mathcal{I}$. Indeed, if $|\mathcal{I}|$ is fixed whilst one takes the number of feature vectors $n \to \infty$ we still expect many properties of GRAND++, in particular Proposition 5 below, to hold. This is due to the asymptotic well-posedness of the dual random walk in low labeling rates [9].*

Proposition 3 reveals that in the simple setting of (2), GRAND converges to a constant when its depth goes to infinity. However, this is not true for GRAND++ since the graph node features will not converge to a constant vector driven by the GRAND++, as shown in Proposition 5 below.

**Proposition 5** *Assume the graph $G = (\boldsymbol{X}, \boldsymbol{W})$ is connected. Then $\boldsymbol{z}^{(i)}(k\delta_t)$ that was defined in (14) does not converge to a constant vector as a function of $i$ as $k \to \infty$. That is, the node features will not become the same across graph nodes under the GRAND++ dynamics.*

**Remark 3** *Proposition 5 guarantees GRAND++ is less likely to suffer from over-smoothing than GRAND, and in particular it shows that we have a non-constant deep layer limit, i.e., as $t \to \infty$. Analysing the limit is beyond the scope of the paper but we have seen one characterisation in Proposition 4. By construction we have $\partial \boldsymbol{z}^{(i)}(t)/\partial t \approx \boldsymbol{0}$ for $i \in \mathcal{I}$ so one should expect that the deep layer limit is (close to) a smooth interpolation of the feature vectors labeled by $\mathcal{I}$.*

The continuous time model (10) is, in the special case of (13), the mean field limit of the probabilistic formulation (17). Our proposed algorithm is formulated from the mean-field limit.

## 6 EXPERIMENTS

In this section, we compare the performance of GRAND++ with GRAND and several other popular GNNs on various graph node classification tasks. We aim to show the practical advantages of GRAND++ in learning with limited labeled data and using deep architectures. Without mentioning clearly, we use the same hyperparameters that that used for GRAND in [10] for GRAND++. We provide detailed descriptions of experimental settings and datasets that are omitted in the main text in Appendix D.1. For all experiments, we run 100 splits for each dataset with 20 random seeds for each split, which are conducted on a server with four NVIDIA RTX 3090 graphics cards.

We compare the performance of GRAND++ and its nonlinear and graph rewiring variants with several popular GNNs on various graph node classification benchmarks. Except for the integration time, which measures the implicit depth of GRAND and GRAND++, we adopt the experimental settings of GRAND in [10] for GRAND++ include numerical differential equation solvers. Following [10], we study seven graph node classification datasets, namely CORA, CiteSeer, PubMed, CoauthorCS, Computer, Photo, and ogbn-arxiv; we describe these datasets in Appendix D.1.

## 6.1 GRAND++ IS MORE RESILIENT TO DEEP ARCHITECTURES

We first show that our introduced source term in (12) can improve the accuracy of GRAND-l when the architecture is deep, i.e., the integration time $T$ in (5) is big. We denote GRAND-l with the source term as GRAND++-l. For each node classification task, we train all models using the same number of labeled nodes as in [10]. Figure 2 contrasts the performance of GRAND-l and GRAND++-l with different depths, or $T$, on CORA, CiteSeer, Computer, and Photo datasets. We provide the detailed results on PubMed and CoauthorCS, together with more comparisons of GRAND++-l with GRAND-l and several other celebrated GNNs include GCN, GAT, and GraphSage in Table 5 in Appendix D.2. The results in Fig. 2 and Appendix D.2 confirm that GRAND-l suffers less from over-smoothing compared to GCN, GAT, and GraphSage. Moreover, GRAND++-l performs on par with GRAND-l when the depth ($T$) of the network is small, but GRAND++-l significantly outperforms GRAND-l when $T$ is large. As $T$ increases, the margin becomes wider, indicating that GRAND++-l can overcome over-smoothing much more effectively than GRAND-l. Note that we did not use uniform depth for GRAND-l and GRAND++-l on all datasets because the adaptive step-size ODE solver fails when $T$ is large for some tasks.

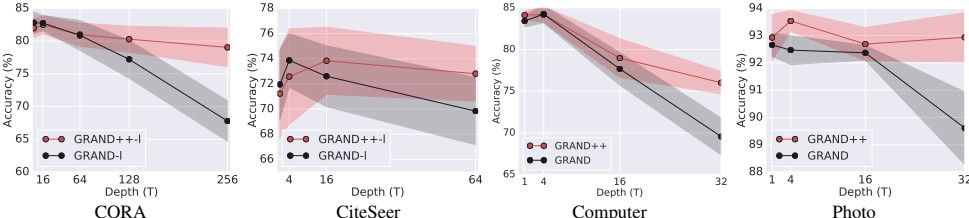

Figure 2: Test accuracy vs. the "depth" ($T$ in (5)) of GRAND-l and GRAND++-l on the four graph node classification tasks. We see that GRAND++-l is much more resilient to deep architectures than GRAND-l. These results show that GRAND++ is better suited for learning with a very deep architecture than GRAND.

Next, we compare GRAND-l and GRAND++-l on the ogbn-arxiv node classification task, which is a large-scale benchmark. We train two models using labeling rates of 3.0% and 5.0%, respectively; the corresponding test accuracy for GRAND-l/GRAND++-l are 65.26%/66.64% and 67.42%/67.77%, respectively. GRAND++-l outperforms GRAND-l in both labeling rates. We further compare GRAND and GRAND++ with different depth on the ogbn-arxiv task in Appendix D.6.

## 6.2 GRAND++ IS MORE ACCURATE WITH LIMITED LABELED TRAINING DATA

Besides helping to overcome over-smoothing, our theory shows that the source term can boost the accuracy of GRAND-l with low-labeling rates. Table 1 compares the accuracy of GRAND++-l with GRAND-l, GCN, GAT, GraphSage, and MoNet, trained with different numbers of labeled data. Here, we slightly tune $T$ for GRAND++ based on the optimal value for GRAND, see Table 4 in the Appendix for their values. We see that with few labeled data, in most tasks GRAND++-l is significantly more accurate than the other GNNs include GRAND-l, confirming our theoretical insight. For Coau-

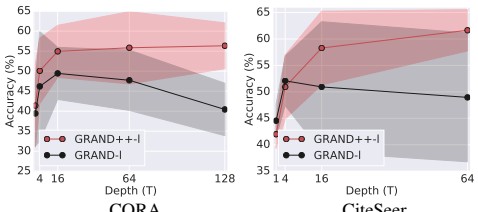

Figure 3: Accuracy of GRAND++-l and GRAND-l for CORA and CiteSeer, where both models, with different depth ($T$), are train with 1 labeled node per class. These results show that GRAND++ is more effective in learning with low-labeling rates than GRAND.

thorCS task, both GRAND-l and GRAND++-l are worse than GCN and GraphSage. Moreover, increasing the depth of GRAND++-l can improve the classification accuracy with limited training data, but this is not the case for GRAND-l, see Fig. 3. We perform the t-test in Appendix D.5 to confirm the statistical significance of the accuracy gain of GRAND++ over GRAND in Table 1.

## 6.3 TIME-DEPENDENT ATTENTION AND GRAPH REWIRING

The previous experimental results show that GRAND++-l enhances the accuracy of GRAND-l in the cases when the labeled training data is limited and when the network is deep. Here, we explore the same strategy for GRAND-nl and GRAND-nl-rw; we name the corresponding models with the new source term GRAND++-nl and GRAND++-nl-rw, respectively. Table 2 compares GRAND-nl and

| Model | #per class | CORA | CiteSeer | PubMed | CoauthorCS | Computer | Photo |
|---|---|---|---|---|---|---|---|
| GRAND++-l (ours) | 1 | **54.94 ± 16.09** | **58.95 ± 9.59** | **65.94 ± 4.87** | 60.30 ± 1.50 | **67.65 ± 0.37** | **83.12 ± 0.78** |
| | 2 | **66.92 ± 10.04** | **64.98 ± 8.31** | **69.31 ± 4.87** | 76.53 ± 1.85 | 76.47 ± 1.48 | **83.71 ± 0.90** |
| | 5 | **77.80 ± 4.46** | **70.03 ± 3.63** | 71.99 ± 1.91 | 84.83 ± 0.84 | **82.64 ± 0.56** | 88.33 ± 1.21 |
| | 10 | **80.86 ± 2.99** | **72.34 ± 2.42** | 75.13 ± 3.88 | 86.94 ± 0.46 | **82.99 ± 0.81** | 90.65 ± 1.19 |
| | 20 | **82.95 ± 1.37** | 73.53 ± 3.31 | **79.16 ± 1.37** | 90.80 ± 0.34 | **85.73 ± 0.50** | **93.55 ± 0.38** |
| GRAND-l [10] | 1 | 52.53 ± 16.40 | 50.06 ± 17.98 | 62.11 ± 10.58 | 59.15 ± 5.73 | 48.67 ± 1.66 | 81.25 ± 2.50 |
| | 2 | 64.82 ± 11.16 | 59.55 ± 10.89 | 69.00 ± 7.55 | 73.83 ± 5.58 | 74.77 ± 1.85 | 82.13 ± 3.27 |
| | 5 | 76.07 ± 5.08 | 68.37 ± 5.00 | **73.98 ± 5.08** | 85.29 ± 2.19 | 80.72 ± 1.09 | 88.27 ± 1.94 |
| | 10 | 80.25 ± 3.40 | 71.90 ± 7.66 | **76.33 ± 3.41** | 87.81 ± 1.36 | 82.42 ± 1.10 | **90.98 ± 0.93** |
| | 20 | 82.86 ± 2.39 | 73.02 ± 5.89 | 78.76 ± 1.69 | 91.03 ± 0.47 | 84.54 ± 0.90 | 93.53 ± 0.47 |
| GCN [30] | 1 | 47.72 ± 15.33 | 48.94 ± 10.24 | 58.61 ± 12.83 | **65.22 ± 2.25** | 49.46 ± 1.65 | 82.94 ± 2.17 |
| | 2 | 60.85 ± 14.01 | 58.06 ± 9.76 | 60.45 ± 16.20 | **83.61 ± 1.49** | **76.90 ± 1.49** | 83.61 ± 0.71 |
| | 5 | 73.86 ± 7.97 | 67.24 ± 4.19 | 68.69 ± 7.93 | 86.66 ± 0.43 | 82.47 ± 0.97 | **88.86 ± 1.56** |
| | 10 | 78.82 ± 5.38 | 72.18 ± 3.47 | 72.59 ± 3.19 | 88.60 ± 0.50 | 82.53 ± 0.74 | 90.41 ± 0.35 |
| | 20 | 82.07 ± 2.03 | **74.21 ± 2.90** | 76.89 ± 3.27 | 91.09 ± 0.35 | 82.94 ± 1.54 | 91.95 ± 0.11 |
| GAT [54] | 1 | 47.86 ± 15.38 | 50.31 ± 14.27 | 58.84 ± 12.81 | 51.13 ± 5.24 | 37.14 ± 7.81 | 73.58 ± 8.15 |
| | 2 | 58.30 ± 13.55 | 55.55 ± 9.19 | 60.24 ± 14.44 | 63.12 ± 6.09 | 65.07 ± 8.86 | 76.89 ± 4.89 |
| | 5 | 71.04 ± 5.74 | 67.37 ± 5.08 | 68.54 ± 5.75 | 71.65 ± 4.53 | 71.43 ± 7.34 | 83.01 ± 3.64 |
| | 10 | 76.31 ± 4.87 | 71.35 ± 4.92 | 72.44 ± 3.50 | 74.71 ± 3.35 | 76.04 ± 0.35 | 87.42 ± 2.38 |
| | 20 | 79.92 ± 2.28 | 73.22 ± 2.90 | 75.55 ± 4.11 | 79.95 ± 2.88 | 80.05 ± 1.81 | 89.38 ± 2.48 |
| GraphSage [29] | 1 | 43.04 ± 14.01 | 48.81 ± 11.45 | 55.53 ± 12.71 | 61.35 ± 1.35 | 27.65 ± 2.39 | 45.36 ± 7.13 |
| | 2 | 53.96 ± 12.18 | 54.39 ± 11.37 | 58.97 ± 12.65 | 76.51 ± 1.31 | 42.63 ± 4.29 | 51.93 ± 4.21 |
| | 5 | 68.14 ± 6.95 | 64.79 ± 5.16 | 66.07 ± 6.16 | **89.06 ± 0.69** | 64.83 ± 1.62 | 78.26 ± 1.93 |
| | 10 | 75.04 ± 5.03 | 68.90 ± 5.08 | 70.74 ± 3.11 | **89.68 ± 0.39** | 74.66 ± 1.29 | 84.38 ± 1.75 |
| | 20 | 80.04 ± 2.54 | 72.02 ± 2.82 | 74.55 ± 3.09 | **91.33 ± 0.36** | 79.98 ± 0.96 | 91.29 ± 0.67 |
| MoNet [40] | 1 | 47.72 ± 15.53 | 39.13 ± 11.37 | 56.47 ± 4.67 | 58.99 ± 5.17 | 23.78 ± 7.57 | 34.72 ± 8.18 |
| | 2 | 60.85 ± 14.01 | 48.52 ± 9.52 | 61.03 ± 6.93 | 76.57 ± 4.06 | 38.19 ± 3.72 | 43.03 ± 8.22 |
| | 5 | 73.86 ± 7.97 | 61.66 ± 6.61 | 67.92 ± 2.50 | 87.02 ± 1.67 | 59.38 ± 4.73 | 71.80 ± 5.02 |
| | 10 | 78.82 ± 5.38 | 68.08 ± 6.29 | 71.24 ± 1.54 | 88.76 ± 0.49 | 68.66 ± 3.30 | 78.66 ± 3.17 |
| | 20 | 82.07 ± 2.03 | 71.52 ± 4.11 | 76.49 ± 1.75 | 90.31 ± 0.41 | 73.66 ± 2.87 | 88.61 ± 1.18 |

Table 1: Classification accuracy of different GNNs trained with different number of labeled data per class (#per class) on six benchmark graph node classification tasks. The highest accuracy is highlighted in bold for each number of labeled data per class. These results show that GRAND++ is more effective in learning with low-labeling rates than GRAND. (Unit: %)

GRAND-nl-rw with the corresponding model with a source term. We see that overall GRAND++-nl (GRAND++-nl-rw) outperforms GRAND-nl (GRAND-nl-rw) when the network is deep, i.e., $T$ is big. We further study the low-labeling rate regimes in Appendix D.3.

| Model | Depth ($T$) | GRAND-nl [10] | GRAND-nl-rw [10] | GRAND++-nl (ours) | GRAND++-nl-rw (ours) |
|---|---|---|---|---|---|
| CORA | 1 | **79.70 ± 1.88** | 79.07 ± 3.05 | 79.24 ± 1.48 | 79.24 ± 1.48 |
| | 4 | 82.31 ± 0.91 | 82.47 ± 1.32 | **82.64 ± 0.89** | 82.23 ± 1.14 |
| | 16 | 82.11 ± 1.42 | 82.05 ± 1.31 | **83.24 ± 0.20** | 81.48 ± 1.07 |
| | 32 | 79.42 ± 0.64 | 81.01 ± 0.81 | 81.21 ± 0.37 | **82.20 ± 1.15** |
| CiteSeer | 1 | 71.84 ± 2.98 | **71.84 ± 2.66** | 70.45 ± 2.12 | 71.74 ± 1.37 |
| | 16 | 72.65 ± 2.42 | 73.06 ± 2.98 | 72.48 ± 1.10 | **73.29 ± 1.37** |
| | 64 | 70.29 ± 2.58 | 69.65 ± 2.50 | 72.64 ± 0.93 | **73.38 ± 0.95** |
| | 128 | 65.19 ± 6.77 | 65.45 ± 7.18 | **74.24 ± 0.70** | 74.23 ± 0.70 |
| PubMed | 1 | 77.93 ± 1.27 | 77.93 ± 1.26 | **78.01 ± 0.68** | 78.01 ± 0.68 |
| | 4 | 77.95 ± 1.28 | 78.02 ± 1.14 | **78.41 ± 0.88** | 78.17 ± 0.93 |
| | 16 | 76.51 ± 2.73 | 76.88 ± 2.57 | **78.43 ± 0.78** | 78.12 ± 0.87 |

Table 2: Classification accuracy of GRAND and GRAND++ variants of different depth trained 20 labeled data per class. The highest accuracy is highlighted in bold for each of the depths $T = 1, 4, 16, 32, 64,$ and $128$. We test $T$ only up to 16 for PubMed and up to 32 for 32 since the neural ODE solver failed for larger $T$. (Unit: %)

## 7 CONCLUDING REMARKS

We propose GRAND++, which augments graph neural diffusion with a source term. We present some theory that connects the model to a random walk formulation on graphs. GRAND++ outperforms many existing GNNs for graph deep learning with very deep architectures and when the number of labeled data is limited. GRAND++ can be regarded as coupled ODE system in which each ODE has an external force term. As such, it is natural to consider if advanced techniques in accelerating training, test, and inference of neural ODEs can be leveraged to improve the efficiency and accuracy of GRAND++, in particular high-order neural ODEs [19, 61, 42, 59] and noise injection [55]. It is interesting to note that the second-order neural ODE can be connected to the wave equation in the graph setting, which can automatically bypass over-smoothing. We leave studying the second-order neural ODE on graphs as future work.

## 8 ACKNOWLEDGEMENT

This material is based on research sponsored by NSF grants DMS-1924935, DMS-1952339, DMS-2027248 and NSF CCF-1934568, DOE grant DE-SC0021142, and ONR grant N00014-18-1-2527 and the MURI grant N00014-20-1-2787. MT would like to thank the Isaac Newton Institute for Mathematical Sciences for support and hospitality during the programme *Mathematics of Deep Learning* when work on this paper was undertaken (EPSRC grant number EP/R014604/1) and acknowledge support from the European Union Horizon 2020 research and innovation programmes under the Marie Skłodowska-Curie grant agreement No. 777826 (NoMADS). MT also holds a Turing Fellowship at the Alan Turing Institute.

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

## A    BACKGROUND ON GRAPH DIFFERENTIAL OPERATORS

Let $(\boldsymbol{X}, \boldsymbol{W})$ represent a graph where $\boldsymbol{X} = ([\boldsymbol{x}^{(1)}]^\top, \ldots, [\boldsymbol{x}^{(n)}]^\top)^\top \in \mathbb{R}^{n \times d}$ is the matrix where each row $\boldsymbol{x}^{(i)} \in \mathbb{R}^d$ is a feature vector and $\boldsymbol{W} = (W_{ij})_{i,j=1}^n$ is a $n \times n$ matrix with $W_{ij}$ representing the similarity (edge weight) between the $i$th and $j$th feature vector. We assume that we are dealing with an undirected graph, i.e., $W_{ij} = W_{ji}$. A $\mathbb{R}^{k_1}$-valued function on the nodes of the graph can be represented as a matrix $\boldsymbol{U} \in \mathbb{R}^{n \times k_1}$ by $\boldsymbol{U} = ([\boldsymbol{u}^{(1)}]^\top, \ldots, [\boldsymbol{u}^{(n)}]^\top)^\top$ and we define the inner product

$$\langle \boldsymbol{U}, \boldsymbol{V} \rangle = \sum_{i=1}^n \boldsymbol{u}^{(i)} \cdot \boldsymbol{v}^{(i)}.$$

Similarly, a $\mathbb{R}^{k_2}$-valued function on the edges can be represented as a third-order tensor $\mathcal{U} \in \mathbb{R}^{n \times n \times k_2}$ which we write as

$$\mathcal{U} = \begin{pmatrix} \mathcal{U}^{(1,1)} & \cdots & \mathcal{U}^{(1,n)} \\ \vdots & \ddots & \vdots \\ \mathcal{U}^{(n,1)} & \cdots & \mathcal{U}^{(n,n)} \end{pmatrix}$$

and $\mathcal{U}^{(i,j)} \in \mathbb{R}^{k_2}$. On edge functions we use the inner product

$$\langle \mathcal{U}, \mathcal{V} \rangle = \frac{1}{2} \sum_{i,j=1}^n W_{ij} \mathcal{U}^{(i,j)} \cdot \mathcal{V}^{(i,j)}.$$

Multiplication between a matrix and an edge function is usually defined pointwise and we use the notation $[\boldsymbol{A} \odot \mathcal{U}]^{(i,j)} = A_{ij} \mathcal{U}^{(i,j)} \in \mathbb{R}^{k_2}$, for a matrix $\boldsymbol{A} \in \mathbb{R}^{n \times n}$ and an edge function $\mathcal{U} \in \mathbb{R}^{n \times n \times k_2}$, to make this clear. Similarly, pointwise multiplication between two matrices $\boldsymbol{A}, \boldsymbol{B} \in \mathbb{R}^{n \times n}$ is defined by $[\boldsymbol{A} \odot \boldsymbol{B}]_{ij} = A_{ij} B_{ij} \in \mathbb{R}$. When a matrix is acting as a linear operator on a node function we use the usual matrix-vector notation and write $[\boldsymbol{A}\boldsymbol{U}]^{(i)} = \sum_{j=1}^n A_{ij} \boldsymbol{u}^{(j)} \in \mathbb{R}^{k_1}$ for a matrix $\boldsymbol{A} \in \mathbb{R}^{n \times n}$ and node function $\boldsymbol{U} = ([\boldsymbol{u}^{(1)}]^\top, \ldots, [\boldsymbol{u}^{(n)}]^\top)^\top \in \mathbb{R}^{n \times k_1}$. In the sequel we will have $k_1 = k_2 = d$.

The gradient of a node-function $\boldsymbol{U} = ([\boldsymbol{u}^{(1)}]^\top, \ldots, [\boldsymbol{u}^{(n)}]^\top)^\top \in \mathbb{R}^{n \times d}$ is defined as the edge-function $\nabla \boldsymbol{U} \in \mathbb{R}^{n \times n \times d}$ with $[\nabla \boldsymbol{U}]^{(i,j)} = \boldsymbol{u}^{(j)} - \boldsymbol{u}^{(i)} \in \mathbb{R}^d$. The divergence $\mathrm{div}\mathcal{V} = ([[\mathrm{div}\mathcal{V}]^{(1)}]^\top, \ldots, [[\mathrm{div}\mathcal{V}]^{(n)}]^\top)^\top \in \mathbb{R}^{n \times d}$ of an edge-function $\mathcal{V} \in \mathbb{R}^{n \times n \times d}$ is defined as

$$[\mathrm{div}\mathcal{V}]^{(i)} = \sum_{j=1}^n W_{ij} \mathcal{V}^{(i,j)}$$

for all $i = 1, \ldots, n$. For anti-symmetric edge functions, i.e. $\mathcal{V}^{(i,j)} = -\mathcal{V}^{(j,i)}$ for all $i, j$, we have that the divergence is the negative adjoint to the gradient, i.e.

$$\langle \mathrm{div}\mathcal{V}, \boldsymbol{U} \rangle = -\langle \mathcal{V}, \nabla \boldsymbol{U} \rangle.$$

## B    TECHNICAL PROOFS

**Proof:**    [Proof of Proposition 1] For notational convenience let us assume that $\boldsymbol{x}^{(i)}(0) = \boldsymbol{x}^{(i)}$. Clearly

$$\mathbb{E}\left[\boldsymbol{B}^{(i)}(0)\right] = \boldsymbol{x}^{(i)} = \boldsymbol{x}^{(i)}(0)$$

for all $i = 1, \ldots, n$. Assume that

$$\mathbb{E}\left[\boldsymbol{B}^{(i)}(k)\right] = \boldsymbol{x}^{(i)}(\delta_t k)$$

for all $i = 1, \ldots, n$. Then,

$$\mathbb{E}\left[\boldsymbol{B}^{(i)}(k+1)\right]$$

$$= \sum_{j=1}^{n} \boldsymbol{x}^{(j)} \mathbb{P}\left(\boldsymbol{B}^{(i)}(k+1) = \boldsymbol{x}^{(j)}\right)$$

$$= \sum_{j=1}^{n} \sum_{\ell=1}^{n} \boldsymbol{x}^{(j)} \mathbb{P}\left(\boldsymbol{B}^{(\ell)}(k) = \boldsymbol{x}^{(j)} | \boldsymbol{B}^{(i)}(1) = \boldsymbol{x}^{(\ell)}\right) \mathbb{P}\left(\boldsymbol{B}^{(i)}(1) = \boldsymbol{x}^{(\ell)}\right)$$

$$= \sum_{j=1}^{n} \sum_{\ell=1}^{n} \boldsymbol{x}^{(j)} \left((1 - \delta_t)\, \mathbb{1}_{i=\ell} + \frac{\delta_t W_{i\ell}}{d_i}\right) \mathbb{P}\left(\boldsymbol{B}^{(i)}(1) = \boldsymbol{x}^{(\ell)}\right)$$

$$= (1 - \delta_t) \sum_{j=1}^{n} \boldsymbol{x}^{(j)} \mathbb{P}\left(\boldsymbol{B}^{(i)}(1) = \boldsymbol{x}^{(\ell)}\right) + \frac{\delta_t}{d_i} \sum_{\ell=1}^{n} W_{i\ell} \sum_{j=1}^{n} \boldsymbol{x}^{(j)} \mathbb{P}\left(\boldsymbol{B}^{(\ell)}(k) = \boldsymbol{x}^{(j)}\right)$$

$$= (1 - \delta_t) \mathbb{E}\left[\boldsymbol{B}^{(i)}(k)\right] + \frac{\delta_t}{d_i} \sum_{\ell=1}^{n} W_{i\ell} \mathbb{E}\left[\boldsymbol{B}^{(\ell)}(k)\right]$$

$$= (1 - \delta_t) \boldsymbol{x}^{(i)}(\delta_t k) + \frac{\delta_t}{d_i} \sum_{\ell=1}^{n} W_{i\ell} \boldsymbol{x}^{(\ell)}(\delta_t k)$$

$$= \boldsymbol{x}^{(i)}(\delta_t k) + \frac{\delta_t}{d_i} \sum_{\ell=1}^{n} W_{i\ell} W_{i\ell} \left(\boldsymbol{x}^{(\ell)}(\delta_t k) - \boldsymbol{x}^{(i)}(\delta_t k)\right)$$

$$= \boldsymbol{x}^{(i)}(\delta_t k) - \delta_t \left[\boldsymbol{L}\boldsymbol{X}(\delta_t k)\right]^{(i)}$$

$$= \boldsymbol{x}^{(i)}(\delta_t(k+1)),$$

as required. $\qquad\square$

**Proof:** [Proof of Proposition 2] Let $\boldsymbol{P} = \left(P_{ij}\right) \in \mathbb{R}^{n \times n}$ be the probability transition kernel, so

$$P_{ij} = \begin{cases} 1 - \delta_t & \text{if } i = j, \\ \frac{\delta_t W_{ij}}{d_i} & \text{if } i \neq j. \end{cases}$$

We have

$$\sum_{i=1}^{n} \pi_i P_{ij} = \sum_{i=1}^{n} \frac{d_i}{\sum_{k=1}^{n} d_k} \left((1 - \delta_t \mathbb{1}_{i=j} + \frac{\delta_t W_{ij}}{d_i}\right)$$

$$= \frac{d_j(1 - \delta_t)}{\sum_{k=1}^{n} d_k} + \frac{\delta_t \sum_{i=1}^{n} W_{ij}}{\sum_{k=1}^{n} d_k}$$

$$= \frac{d_j}{\sum_{k=1}^{n} d_k}$$

$$= \pi_j,$$

as required. $\qquad\square$

**Proof:** [Proof of Proposition 3] The proof follows from a simple application of Propositions 1 and 2. Namely, for any $i \in \{1, \ldots, n\}$

$$\boldsymbol{x}^{(i)}(k\delta_t) = \mathbb{E}\left[\boldsymbol{B}^{(i)}(k)\right] = \sum_{j=1}^{n} \boldsymbol{x}^{(j)}(0) \mathbb{P}\left(\boldsymbol{B}^{(i)}(k) = \boldsymbol{x}^{(j)}\right) \to \sum_{j=1}^{n} \boldsymbol{x}^{(j)}(0) \pi_j = \widetilde{\boldsymbol{x}}$$

as $k \to \infty$. $\qquad\square$

**Proof:** [Proof of Proposition 4] Let

$$\boldsymbol{y}^{(i)}(k) = \mathbb{E}\left[\sum_{s=0}^{k} \frac{1}{d_i} \sum_{j \in \mathcal{I}} \left(\boldsymbol{x}^{(j)} - \hat{\boldsymbol{x}}\right) \mathbb{1}_{\boldsymbol{B}^{(j)}(s) = \boldsymbol{x}^{(i)}}\right].$$

Notice that

$$
\mathbb{E}\left[\sum_{s=0}^{k} \mathbb{1}_{\boldsymbol{B}^{(j)}(s)=\boldsymbol{x}^{(i)}}\right]
$$

$$
= \sum_{s=0}^{k} \mathbb{P}\left(\boldsymbol{B}^{(j)}(s) = \boldsymbol{x}^{(i)}\right)
$$

$$
= \underbrace{\mathbb{P}\left(\boldsymbol{B}^{(j)}(0) = \boldsymbol{x}^{(i)}\right)}_{\delta_{ij}} + \sum_{s=1}^{k} \mathbb{P}\left(\boldsymbol{B}^{(j)}(s) = \boldsymbol{x}^{(i)}\right)
$$

$$
= \delta_{ij} + \sum_{s=1}^{k}\sum_{\ell=1}^{n} \mathbb{P}\left(\boldsymbol{B}^{(j)}(s) = \boldsymbol{x}^{(i)}|\boldsymbol{B}^{(j)}(s-1) = \boldsymbol{x}^{(\ell)}\right)\mathbb{P}\left(\boldsymbol{B}^{(j)}(s-1) = \boldsymbol{x}^{(\ell)}\right)
$$

$$
= \delta_{ij} + \sum_{s=1}^{k}\sum_{\ell=1}^{n}\left((1-\delta_t)\delta_{\ell i} + \frac{\delta_t W_{\ell i}}{d_\ell}\right)\mathbb{P}\left(\boldsymbol{B}^{(j)}(s-1) = \boldsymbol{x}^{(\ell)}\right)
$$

$$
= \delta_{ij} + (1-\delta_t)\sum_{s=1}^{k}\mathbb{P}\left(\boldsymbol{B}^{(j)}(s-1) = \boldsymbol{x}^{(i)}\right) + \delta_t\sum_{\ell=1}^{n}\frac{W_{\ell i}}{d_\ell}\sum_{s=1}^{k}\mathbb{P}\left(\boldsymbol{B}^{(j)}(s-1) = \boldsymbol{x}^{(\ell)}\right)
$$

$$
= \delta_{ij} + (1-\delta_t)\sum_{s=0}^{k-1}\mathbb{P}\left(\boldsymbol{B}^{(j)}(s) = \boldsymbol{x}^{(i)}\right) + \delta_t\sum_{\ell=1}^{n}\frac{W_{\ell i}}{d_\ell}\sum_{s=0}^{k-1}\mathbb{P}\left(\boldsymbol{B}^{(j)}(s) = \boldsymbol{x}^{(\ell)}\right)
$$

$$
= \delta_{ij} + (1-\delta_t)\mathbb{E}\left[\sum_{s=0}^{k-1}\mathbb{1}_{\boldsymbol{B}^{(j)}(s)=\boldsymbol{x}^{(i)}}\right] + \delta_t\sum_{\ell=1}^{n}\frac{W_{\ell i}}{d_\ell}\mathbb{E}\left[\sum_{s=0}^{k-1}\mathbb{1}_{\boldsymbol{B}^{(j)}(s)=\boldsymbol{x}^{(\ell)}}\right].
$$

From the definition of $\boldsymbol{Y}$ and the above recursive relationship we have

$$
\boldsymbol{y}^{(i)}(k) = \frac{1}{d_i}\sum_{j\in\mathcal{I}}\left(\boldsymbol{x}^{(j)} - \hat{\boldsymbol{x}}\right)\mathbb{E}\left[\sum_{s=0}^{k}\mathbb{1}_{\boldsymbol{B}^{(j)}(s)=\boldsymbol{x}^{(i)}}\right]
$$

$$
= \frac{1}{d_i}\sum_{j\in\mathcal{I}}\left(\boldsymbol{x}^{(j)} - \hat{\boldsymbol{x}}\right)\delta_{ij} + (1-\delta_t)\frac{1}{d_i}\sum_{j\in\mathcal{I}}\left(\boldsymbol{x}^{(j)} - \hat{\boldsymbol{x}}\right)\mathbb{E}\left[\sum_{s=0}^{k-1}\mathbb{1}_{\boldsymbol{B}^{(j)}(s)=\boldsymbol{x}^{(\ell)}}\right]
$$

$$
+ \frac{\delta_t}{d_i}\sum_{\ell=1}^{n}\frac{W_{\ell i}}{d_\ell}\sum_{j\in\mathcal{I}}\left(\boldsymbol{x}^{(j)} - \hat{\boldsymbol{x}}\right)\mathbb{E}\left[\sum_{s=0}^{k-1}\mathbb{1}_{\boldsymbol{B}^{(j)}(s)=\boldsymbol{x}^{(\ell)}}\right]
$$

$$
= \frac{1}{d_i}\sum_{j\in\mathcal{I}}\left(\boldsymbol{x}^{(j)} - \hat{\boldsymbol{x}}\right)\delta_{ij} + (1-\delta_t)\boldsymbol{y}^{(i)}(k-1) + \frac{\delta_t}{d_i}\sum_{\ell=1}^{n}W_{\ell i}\boldsymbol{y}^{(\ell)}(k-1)
$$

$$
= \boldsymbol{y}^{(i)}(k-1) + \frac{1}{d_i}\sum_{j\in\mathcal{I}}\left(\boldsymbol{x}^{(j)} - \hat{\boldsymbol{x}}\right)\delta_{ij} - \delta_t\left[\boldsymbol{L}\boldsymbol{Y}(k-1)\right]^{(i)}.
$$

Now we let

$$
\boldsymbol{w}^{(i)}(k) = d_i\left(\boldsymbol{z}^{(i)}(k\delta_t) - \boldsymbol{y}^{(i)}(k)\right)
$$

so that $\boldsymbol{W}$ satisfies

$$
\boldsymbol{w}^{(i)}(k) = \boldsymbol{w}^{(i)}(k-1) - \delta_t\left[\boldsymbol{L}\boldsymbol{W}(k-1)\right]^{(i)} = \left[\boldsymbol{P}\boldsymbol{W}(k-1)\right]^{(i)}.
$$

Hence, $\boldsymbol{W}(k) = \boldsymbol{P}^k\boldsymbol{W}(0)$. Since the stationary distribution of the random walk with transition kernel is $\pi$, we have $\lim_{k\to\infty}\boldsymbol{P}^k = \mathbb{1}\pi^\top$. Hence, as $k \to \infty$, $\boldsymbol{w}^{(i)}(k) \to \sum_{j=1}^{n}\pi_j\boldsymbol{w}^{(j)}(0) = 0$ since $\boldsymbol{w}^{(j)}(0) = 0$ by the choice in the initialisation of $\boldsymbol{Z}$. $\qquad\square$

**Proof:** [Proof of Proposition 5] We prove the result by contradiction. Assume that there exists $\overline{\boldsymbol{z}}$ such that $\boldsymbol{z}^{(i)}(k\delta_t) \to \overline{\boldsymbol{z}}$ for all $i = 1, \ldots, n$ as $k \to \infty$. Then $\boldsymbol{LZ}(k\delta_t) \to 0$. Since we can write

$$\boldsymbol{z}^{(i)}(k\delta_t) - \boldsymbol{z}^{(j)}(k\delta_t) = \boldsymbol{z}^{(i)}((k-1)\delta_t) - \boldsymbol{z}^{(j)}((k-1)\delta_t)$$
$$- \delta_t \left( [\boldsymbol{LZ}((k-1)\delta_t)]^{(i)} - [\boldsymbol{LZ}((k-1)\delta_t)]^{(j)} \right)$$
$$+ \delta_t \sum_{\ell \in \mathcal{I}} \left( \delta_{i\ell}(\boldsymbol{x}^{(i)} - \hat{\boldsymbol{x}}) - \delta_{j\ell}(\boldsymbol{x}^{(j)} - \hat{\boldsymbol{x}}) \right),$$

then taking the limit $k \to \infty$ implies

$$\sum_{\ell \in \mathcal{I}} \delta_{i\ell}(\boldsymbol{x}^{(i)} - \hat{\boldsymbol{x}}) = \sum_{\ell \in \mathcal{I}} \delta_{j\ell}(\boldsymbol{x}^{(j)} - \hat{\boldsymbol{x}})$$

for all $i, j$ which is clearly not true. $\qquad\square$

## C  NEURAL ODEs AND TRAINING NEURAL ODEs WITH ADJOINT METHOD

Neural ODEs [13] are a class of continuous-depth (-time) neural networks that are particularly suitable for learning complex dynamics from irregularly sampled sequential data, see, e.g., [13, 51, 19, 39, 42]. Mathematically, a neural ODE is the first-order ODE:

$$\frac{d\boldsymbol{h}(t)}{dt} = f(\boldsymbol{h}(t), t, \theta), \tag{18}$$

where $f(\boldsymbol{h}(t), t, \theta) \in \mathbb{R}^d$ is specified by a neural network parameterised by $\theta$, e.g., a two-layer feedforward neural network. Starting from the input $\boldsymbol{h}(0)$, neural ODEs learn the representation and perform prediction by solving (18) from $t = 0$ to $T$ using a numerical integrator with a given error tolerance, often with an adaptive step size solver (or adaptive solver for short) [18]. Solving (18) from $t = 0$ to $T$ in a single pass with an adaptive solver requires evaluating $f(\boldsymbol{h}(t), t, \theta)$ at various timesteps, with computational complexity counted by the number of forward function evaluations (forward NFEs) [13].

The adjoint sensitivity method (or adjoint method) [46], is a memory-efficient method for training neural ODEs. We regard the output $\boldsymbol{h}(T)$ as the prediction and denote the loss between $\boldsymbol{h}(T)$ and the ground truth as $\mathcal{L}$. Let $\boldsymbol{a}(t) := \partial\mathcal{L}/\partial\boldsymbol{h}(t)$ be the adjoint state, then we have (see [13, 46] for details)

$$\frac{d\mathcal{L}}{d\theta} = \int_0^T \boldsymbol{a}(t)^\top \frac{\partial f(\boldsymbol{h}(t), t, \theta)}{\partial \theta} dt, \tag{19}$$

with $\boldsymbol{a}(t)$ satisfying the following adjoint ODE

$$\frac{d\boldsymbol{a}(t)}{dt} = -\boldsymbol{a}(t)^\top \frac{\partial}{\partial \boldsymbol{h}} f(\boldsymbol{h}(t), t, \theta), \tag{20}$$

which is solved numerically from $t = T$ to $0$ and also requires the evaluation of the right-hand side of (20) at various timestamps, and the backward NFEs measure the computational complexity.

## D  EXPERIMENTAL DETAILS AND MORE EXPERIMENTAL RESULTS

### D.1  DATASETS AND EXPERIMENTAL SETTINGS

**Graph node classification dataset.** Following [10], we consider the largest connected component of seven graph node classification datasets, including CORA, CiteSeer, PubMed, coauthor graph CoauthorCS, and Amazon co-purchasing graphs Computer and Photo, and a large scale ogbn-arxiv dataset. For completeness, we list the number of classes, the number of features, and the number of nodes and edges of each dataset in Table 3. More detailed information can be found in [10].

**Depth of GRAND and GRAND++ for the results in Table 1.** Table 4 lists the fine-tuned $T$ for the results in Table 1. Due to the limited time, we only search around the value of optimal $T$ for GRAND with grid spacing 0.1.

| Dataset | Classes | Features | #Nodes | #Edges |
|---------|---------|----------|--------|--------|
| CORA | 7 | 1433 | 2485 | 5069 |
| CiteSeer | 6 | 3703 | 2120 | 3679 |
| PubMed | 3 | 500 | 19717 | 44324 |
| CoauthorCS | 15 | 6805 | 18333 | 81894 |
| Computer | 10 | 767 | 13381 | 245778 |
| Photo | 8 | 745 | 7487 | 119043 |
| ogbn-arxiv | 40 | 128 | 169343 | 1166243 |

Table 3: Summary of the graph node classification datasets.

| Model | CORA | CiteSeer | PubMed | CoauthorCS | Computer | Photo |
|-------|------|----------|--------|------------|----------|-------|
| GRAND++-l | 18.3 | 8.0 | 13.0 | 4.0 | 3.2 | 3.6 |
| GRAND-l | 18.2948 | 7.8741 | 12.9423 | 3.2490 | 3.5824 | 3.6760 |

Table 4: The value of the fine-tuned $T$, i.e. depth of the continuous-depth GNNs, for GRAND and GRAND++ in learning with different labeling results, and the corresponding accuracy are reported in Table 1. The values of $T$ for GRAND++ are adopted from the paper [10].

## D.2 CLASSIFICATION ACCURACY OF GNNS WITH DIFFERENT DEPTHS

In this subsection, we provide detailed numbers that correspond to Fig. 1. We further compare GRAND++-l with several other GNN architectures, including GCN, GAT, and GraphSage, with different depths on a few benchmark datasets. Table 5 lists the classification accuracy of GRAND++, GRAND, and three benchmark GNNs with different depths on six graph node classification tasks. Again, we see that GRAND++ is better than the other GNN models when the networks are deep.

| Model | depth | CORA | CiteSeer | PubMed | CoauthorCS | Computer | Photo |
|-------|-------|------|----------|--------|------------|----------|-------|
| GRAND++-l (ours) | 1 | $77.48 \pm 1.43$ | $71.23 \pm 3.47$ | $\mathbf{78.11 \pm 1.47}$ | $90.42 \pm 0.76$ | $\mathbf{84.11 \pm 0.51}$ | $\mathbf{92.93 \pm 0.84}$ |
| | 4 | $81.98 \pm 1.42$ | $72.58 \pm 3.79$ | $\mathbf{79.20 \pm 0.74}$ | $90.89 \pm 0.36$ | $84.19 \pm 0.93$ | $\mathbf{93.54 \pm 0.38}$ |
| | 16 | $82.49 \pm 1.37$ | $\mathbf{73.84 \pm 2.66}$ | $79.49 \pm 0.84$ | $\mathbf{90.24 \pm 0.30}$ | $\mathbf{78.97 \pm 2.33}$ | $92.69 \pm 0.61$ |
| | 32 | $82.48 \pm 0.71$ | $73.29 \pm 1.29$ | $\mathbf{79.81 \pm 1.61}$ | NA | $\mathbf{76.01 \pm 1.33}$ | $\mathbf{92.94 \pm 0.90}$ |
| | 64 | $\mathbf{80.99 \pm 1.76}$ | $\mathbf{72.81 \pm 2.18}$ | NA | NA | NA | NA |
| | 128 | $\mathbf{80.29 \pm 1.98}$ | NA | NA | NA | NA | NA |
| | 256 | $\mathbf{79.04 \pm 2.94}$ | NA | NA | NA | NA | NA |
| GRAND-l [10] | 1 | $\mathbf{78.59 \pm 1.17}$ | $71.96 \pm 2.74$ | $77.93 \pm 1.26$ | $90.79 \pm 0.93$ | $83.41 \pm 0.69$ | $92.66 \pm 0.42$ |
| | 4 | $\mathbf{82.80 \pm 1.62}$ | $\mathbf{73.87 \pm 2.12}$ | $78.71 \pm 1.19$ | $\mathbf{90.94 \pm 0.21}$ | $\mathbf{84.23 \pm 1.05}$ | $92.47 \pm 0.53$ |
| | 16 | $\mathbf{82.75 \pm 1.17}$ | $72.61 \pm 2.42$ | $78.79 \pm 0.93$ | $87.66 \pm 1.70$ | $77.67 \pm 1.94$ | $92.37 \pm 0.27$ |
| | 32 | $82.19 \pm 1.73$ | $72.65 \pm 3.15$ | $78.70 \pm 1.08$ | NA | $69.56 \pm 2.20$ | $89.61 \pm 1.33$ |
| | 64 | $80.87 \pm 2.28$ | $69.84 \pm 2.66$ | NA | NA | NA | NA |
| | 128 | $77.22 \pm 2.88$ | NA | NA | NA | NA | NA |
| | 256 | $67.79 \pm 3.10$ | NA | NA | NA | NA | NA |
| GCN [30] | 1 | $76.92 \pm 0.56$ | $\mathbf{72.80 \pm 1.69}$ | $72.78 \pm 1.80$ | $91.53 \pm 0.45$ | $81.44 \pm 0.24$ | $91.31 \pm 0.19$ |
| | 4 | $81.35 \pm 1.27$ | $70.54 \pm 6.61$ | $77.15 \pm 3.00$ | $87.84 \pm 0.96$ | $75.73 \pm 1.02$ | $90.11 \pm 0.66$ |
| | 16 | $19.70 \pm 7.06$ | $24.78 \pm 1.45$ | $41.36 \pm 1.77$ | $14.49 \pm 0.91$ | $12.86 \pm 2.39$ | $23.11 \pm 1.76$ |
| | 32 | $21.86 \pm 6.09$ | $24.23 \pm 1.65$ | $40.66 \pm 1.86$ | $12.14 \pm 1.64$ | $21.15 \pm 13.10$ | $24.30 \pm 0.73$ |
| GAT [54] | 1 | $72.49 \pm 2.03$ | $71.83 \pm 1.53$ | $77.24 \pm 0.72$ | $79.22 \pm 0.60$ | $73.97 \pm 1.20$ | $87.08 \pm 0.37$ |
| | 4 | $80.95 \pm 2.28$ | $72.31 \pm 2.82$ | $77.37 \pm 1.32$ | $78.05 \pm 1.10$ | $76.67 \pm 2.79$ | $87.95 \pm 1.76$ |
| | 16 | $29.14 \pm 1.02$ | $24.84 \pm 1.45$ | $39.21 \pm 0.43$ | $24.20 \pm 2.22$ | $37.07 \pm 2.99$ | $29.97 \pm 3.68$ |
| | 32 | $29.75 \pm 1.57$ | $24.83 \pm 1.45$ | $39.02 \pm 0.12$ | $\mathbf{22.73 \pm 2.08}$ | $32.53 \pm 3.09$ | $25.57 \pm 4.03$ |
| GraphSage [29] | 1 | $73.47 \pm 1.98$ | $71.94 \pm 1.45$ | $72.42 \pm 0.61$ | $\mathbf{91.74 \pm 0.26}$ | $75.95 \pm 0.70$ | $88.10 \pm 0.87$ |
| | 4 | $79.83 \pm 2.43$ | $50.00 \pm 14.27$ | $76.01 \pm 2.35$ | $87.94 \pm 2.35$ | $75.62 \pm 2.85$ | $90.68 \pm 2.11$ |
| | 16 | $25.52 \pm 6.45$ | $24.84 \pm 1.45$ | $37.55 \pm 3.92$ | $10.12 \pm 2.21$ | $22.79 \pm 10.77$ | $25.57 \pm 3.31$ |
| | 32 | $29.14 \pm 1.02$ | $28.38 \pm 2.54$ | $39.21 \pm 4.39$ | $7.91 \pm 3.15$ | $37.07 \pm 13.22$ | $20.09 \pm 5.67$ |

Table 5: Classification accuracy of different GNN models with different depths on six benchmark graph node classification tasks. NA: neural ODE solver failed. These results show that GRAND++ is better suited for learning with a very deep architecture than GRAND. (Unit: %)

## D.3 MORE RESULTS ON TIME-DEPENDENT ATTENTION AND GRAPH REWIRING

We further explore the effects of the source term for GRAND-nl and GRAND-nl-rw in the low-labeling rate regimes. Table 6 compares GRAND-nl and GRAND-nl-rw with the corresponding model with a source term. We see that GRAND-nl and GRAND-nl-rw are almost always worse than the vanilla GRAND-l, consistent with the results reported in [10]. GRAND++-nl and GRAND++-nl-rw cannot help learning at low labeling rates anymore. However, when the labeling rates are not low, GRAND++-nl or GRAND++-nl-rw can outperform GRAND-nl and GRAND-nl-rw, even outperform GRAND++.

| Model | #per class | GRAND-nl [10] | GRAND-nl-rw [10] | GRAND++-nl (ours) | GRAND++-nl-rw (ours) |
|---|---|---|---|---|---|
| CORA | 1 | $50.55 \pm 15.68$ | $50.63 \pm 17.71$ | $48.89 \pm 11.51$ | $47.94 \pm 11.06$ |
| | 2 | $65.06 \pm 9.35$ | $61.24 \pm 16.19$ | $59.96 \pm 7.90$ | $58.25 \pm 11.97$ |
| | 5 | $76.93 \pm 3.10$ | $76.50 \pm 3.91$ | $74.01 \pm 1.73$ | $74.25 \pm 1.99$ |
| | 10 | $79.60 \pm 2.69$ | $79.38 \pm 3.25$ | $80.14 \pm 0.69$ | $80.18 \pm 0.40$ |
| | 20 | $82.22 \pm 1.93$ | $82.14 \pm 2.49$ | $83.24 \pm 0.20$ | $81.48 \pm 1.07$ |
| CiteSeer | 1 | $50.25 \pm 17.66$ | $50.20 \pm 17.90$ | $49.65 \pm 5.45$ | $53.10 \pm 5.51$ |
| | 2 | $59.87 \pm 10.89$ | $59.95 \pm 10.48$ | $59.16 \pm 8.13$ | $60.26 \pm 5.10$ |
| | 5 | $68.21 \pm 5.08$ | $68.05 \pm 5.48$ | $66.13 \pm 2.09$ | $67.81 \pm 1.97$ |
| | 10 | $71.88 \pm 6.94$ | $71.92 \pm 7.34$ | $68.84 \pm 2.84$ | $71.45 \pm 1.64$ |
| | 20 | $72.84 \pm 6.61$ | $72.72 \pm 6.85$ | $72.52 \pm 1.24$ | $73.87 \pm 1.35$ |
| PubMed | 1 | $66.97 \pm 10.07$ | $67.69 \pm 7.89$ | $63.85 \pm 4.86$ | $67.45 \pm 3.88$ |
| | 2 | $69.17 \pm 2.46$ | $69.42 \pm 2.13$ | $66.98 \pm 5.30$ | $69.11 \pm 1.80$ |
| | 5 | $72.56 \pm 3.36$ | $72.68 \pm 2.52$ | $71.49 \pm 1.53$ | $72.05 \pm 3.67$ |
| | 10 | $76.03 \pm 3.73$ | $75.32 \pm 3.45$ | $74.94 \pm 2.15$ | $75.09 \pm 2.88$ |
| | 20 | $78.55 \pm 1.59$ | $78.30 \pm 1.43$ | $78.41 \pm 0.99$ | $79.44 \pm 0.56$ |

Table 6: Classification accuracy of the variants of GRAND and GRAND++ models trained with different numbers of labeled data per class (#per class) on graph node classification tasks. (Unit: %)

## D.4 CLASSIFICATION ACCURACY OF GNNs WITH FEWER NUMBER OF TRAINING DATA

Besides the results shown in Fig. 1, we further test the classification accuracy of more benchmark GNN architectures trained with fewer numbers of labeled data per class. Tables 7-9 list the classification accuracy, on the test set, of different benchmark GNN models when they are trained with different numbers of labeled nodes per class.

| #labeled nodes per class | 1 | 2 | 5 | 10 | 20 |
|---|---|---|---|---|---|
| GCN [30] | $47.72 \pm 15.53$ | $60.85 \pm 14.01$ | $73.86 \pm 7.97$ | $78.82 \pm 5.38$ | $82.07 \pm 2.03$ |
| GAT [54] | $47.86 \pm 15.38$ | $58.30 \pm 13.55$ | $71.04 \pm 5.74$ | $76.31 \pm 4.87$ | $79.92 \pm 2.28$ |
| GraphSage [29] | $43.04 \pm 14.01$ | $53.96 \pm 12.18$ | $68.14 \pm 6.95$ | $75.04 \pm 5.03$ | $80.04 \pm 2.54$ |
| MoNet [40] | $47.72 \pm 15.53$ | $60.85 \pm 14.01$ | $73.86 \pm 7.97$ | $78.82 \pm 5.38$ | $82.07 \pm 2.03$ |
| Lanczos [36] | $47.41 \pm 11.82$ | $60.94 \pm 4.00$ | $74.28 \pm 3.07$ | $76.12 \pm 0.93$ | $79.85 \pm 1.82$ |
| AdaLanczos [36] | $48.23 \pm 11.82$ | $61.46 \pm 4.96$ | $74.24 \pm 3.25$ | $77.61 \pm 1.36$ | $81.03 \pm 1.56$ |
| GCNN [58] | $43.31 \pm 11.95$ | $60.28 \pm 12.89$ | $72.75 \pm 4.21$ | $78.92 \pm 1.32$ | $81.89 \pm 1.12$ |
| GRAND-l [10] | $52.53 \pm 16.40$ | $64.82 \pm 11.16$ | $76.07 \pm 5.08$ | $80.25 \pm 3.40$ | $82.86 \pm 2.39$ |
| GRAND-nl [10] | $40.97 \pm 14.87$ | $50.59 \pm 13.25$ | $65.13 \pm 9.14$ | $72.55 \pm 6.65$ | $77.76 \pm 4.21$ |
| GRAND-nl-rw (gdc) [10] | $52.68 \pm 12.48$ | $65.54 \pm 10.01$ | $74.94 \pm 7.04$ | $80.64 \pm 6.19$ | $82.47 \pm 1.93$ |
| GRAND-nl-rw (two-hop) [10] | $53.79 \pm 17.72$ | $64.50 \pm 11.88$ | $74.33 \pm 6.28$ | $79.61 \pm 4.47$ | $82.37 \pm 1.98$ |

Table 7: Classification accuracy of different GNNs trained with different numbers of labeled nodes per class. Dataset: CORA.

| #labeled nodes per class | 1 | 2 | 5 | 10 | 20 |
|---|---|---|---|---|---|
| GCN [30] | $48.94 \pm 10.24$ | $58.06 \pm 9.76$ | $67.24 \pm 4.19$ | $72.18 \pm 3.47$ | $74.21 \pm 2.90$ |
| GAT [54] | $50.31 \pm 14.27$ | $55.55 \pm 9.19$ | $67.37 \pm 5.08$ | $71.35 \pm 4.92$ | $73.22 \pm 2.90$ |
| GraphSage [29] | $48.81 \pm 11.45$ | $54.39 \pm 11.37$ | $64.79 \pm 5.16$ | $68.90 \pm 5.08$ | $72.02 \pm 2.82$ |
| MoNet [40] | $39.13 \pm 11.37$ | $48.52 \pm 9.52$ | $61.66 \pm 6.61$ | $68.08 \pm 6.29$ | $71.52 \pm 4.11$ |
| Lanczos [36] | $49.16 \pm 3.63$ | $57.65 \pm 7.60$ | $66.72 \pm 9.38$ | $71.01 \pm 4.90$ | $72.14 \pm 2.00$ |
| AdaLanczos [36] | $50.32 \pm 7.42$ | $58.35 \pm 7.97$ | $67.39 \pm 8.20$ | $72.15 \pm 4.85$ | $74.33 \pm 2.83$ |
| GCNN [58] | $40.58 \pm 15.32$ | $51.71 \pm 13.87$ | $63.16 \pm 12.26$ | $67.06 \pm 5.65$ | $69.84 \pm 1.77$ |
| GRAND-l [10] | $50.06 \pm 17.98$ | $59.55 \pm 10.89$ | $68.37 \pm 5.00$ | $71.90 \pm 7.66$ | $73.02 \pm 5.89$ |
| GRAND-nl [10] | $49.96 \pm 18.62$ | $59.57 \pm 11.03$ | $68.21 \pm 7.08$ | $71.88 \pm 6.94$ | $72.84 \pm 6.61$ |
| GRAND-nl-rw (gdc) [10] | $50.35 \pm 17.74$ | $59.98 \pm 10.32$ | $68.39 \pm 5.81$ | $71.83 \pm 7.26$ | $72.81 \pm 6.94$ |
| GRAND-nl-rw (two-hop) [10] | $50.20 \pm 17.90$ | $59.95 \pm 10.48$ | $68.05 \pm 5.49$ | $71.92 \pm 7.34$ | $72.72 \pm 6.85$ |

Table 8: Classification accuracy of different GNNs trained with different numbers of labeled nodes per class. Dataset: CiteSeer.

## D.5 T-TEST OF THE ACCURACY IMPROVEMENT OF GRAND++ OVER GRAND

To confirm the statistical significance of the accuracy improvement of GRAND++ over GRAND in Table 1, in this subsection, we conduct t-test experiments at 0.95 confidence to compare GRAND and GRAND++ on six different benchmark graph node classification tasks. We first perform unpaired t-tests to show the improvement of GRAND++ over GRAND on low labeled datasets using the following t-score

$$\textbf{t-score} = \frac{\mu_{\textbf{GRAND++}} - \mu_{\textbf{GRAND}}}{\sqrt{\frac{\sigma^2_{\textbf{GRAND++}}}{n} + \frac{\sigma^2_{\textbf{GRAND++}}}{n}}}, \tag{21}$$

| #labeled nodes per class | 1 | 2 | 5 | 10 | 20 |
|---|---|---|---|---|---|
| GCN [30] | $58.61 \pm 12.83$ | $60.45 \pm 16.20$ | $68.69 \pm 7.93$ | $72.59 \pm 3.19$ | $76.89 \pm 3.27$ |
| GAT [54] | $58.84 \pm 12.81$ | $60.24 \pm 14.44$ | $68.54 \pm 5.75$ | $72.44 \pm 3.50$ | $75.55 \pm 4.11$ |
| GraphSage [29] | $55.53 \pm 12.71$ | $58.97 \pm 12.65$ | $66.07 \pm 6.16$ | $70.74 \pm 3.11$ | $74.55 \pm 3.09$ |
| MoNet [40] | $56.47 \pm 4.67$ | $61.03 \pm 6.93$ | $67.92 \pm 2.50$ | $71.24 \pm 1.54$ | $76.49 \pm 1.75$ |
| Lanczos [36] | $60.12 \pm 6.37$ | $63.65 \pm 6.97$ | $70.61 \pm 4.50$ | $73.01 \pm 3.27$ | $78.35 \pm 1.84$ |
| AdaLanczos [36] | $61.07 \pm 5.16$ | $64.11 \pm 6.88$ | $69.05 \pm 3.00$ | $72.79 \pm 2.74$ | $78.10 \pm 1.91$ |
| GCNN [58] | $60.78 \pm 20.64$ | $65.14 \pm 19.45$ | $72.72 \pm 10.82$ | $76.47 \pm 6.03$ | $79.24 \pm 3.45$ |
| GRAND-l [10] | $62.11 \pm 10.58$ | $69.00 \pm 7.55$ | $73.98 \pm 5.08$ | $76.33 \pm 3.41$ | $78.76 \pm 1.69$ |
| GRAND-nl [10] | $61.75 \pm 11.12$ | $69.16 \pm 8.46$ | $72.35 \pm 5.35$ | $76.03 \pm 3.72$ | $78.55 \pm 1.59$ |
| GRAND-nl-rw (gdc) [10] | $61.70 \pm 10.74$ | $69.42 \pm 8.21$ | $72.39 \pm 5.25$ | $75.32 \pm 3.45$ | $78.30 \pm 1.43$ |
| GRAND-nl-rw (two-hop) [10] | $61.65 \pm 12.09$ | $68.49 \pm 8.99$ | $72.68 \pm 5.92$ | $75.72 \pm 3.50$ | $78.77 \pm 1.88$ |

Table 9: Classification accuracy of different GNNs trained with different numbers of labeled nodes per class. Dataset: PubMed.

where $\mu$ and $\sigma^2$ are the mean and variance of the performances of each model, and $n$ is the number of runs for each model. The t-test score are shown in Table 10.

| #per class | CORA | CiteSeer | PubMed | CoauthorCS | Computer | Photo |
|---|---|---|---|---|---|---|
| 1 | 1.05 | **4.36** | **3.28** | **1.95** | **111.60** | **45.33** |
| 2 | 1.39 | **3.96** | 0.34 | **4.59** | **7.18** | **4.65** |
| 5 | **2.55** | **2.68** | -3.67 | -1.96 | **15.67** | 0.26 |

Table 10: Unpaired t-test scores of GRAND++ v.s. GRAND on six different benchmark graph node classification tasks. With $n = 100$, over 0.95 confidence is equivalent to exceed roughly 1.66 t-test scores. Highlighted are the ones passing the test.

For some entries in Table 10 that are not significant enough, we further conduct paired t-test between GRAND++ and GRAND on these specific datasets as shown in Table 11. Since a large portion of variance comes from splitting of the datasets, we pair up tests of GRAND and GRAND++ with the same splitting in this experiment. In this case, a sample of difference of size $n$ is computed, and t-test score can be computed using the equation

$$\mathbf{t\text{-}score} = \frac{\mu_{\mathbf{diff}} - 0}{\sigma_{\mathbf{diff}}/\sqrt{n}}. \tag{22}$$

| Dataset | #per class | Accuracy Difference | # splits | t-score | p-score |
|---|---|---|---|---|---|
| CORA | 1 | $1.06 \pm 6.24$ | 100 | 1.80 | 0.044 |
| CORA | 2 | $1.45 \pm 5.23$ | 100 | 2.78 | 0.003 |

Table 11: Paired t-test scores of GRAND++ v.s. GRAND on datasets where unpaired t-test scores are not significant enough.

## D.6 TASKS FOR FURTHER EVALUATING DEEP GRAND AND GRAND++

**Open graph benchmark with paper citation network (ogbn-arxiv).** Ogbn-arxiv consists of 169, 343 nodes and 1, 166, 243 directed edges. Each node is an arxiv paper represented by a 128-dimensional features and each directed edge indicates the citation direction. This dataset is used for node property prediction and has been a popular benchmark to test the advantage of deep graph neural networks over shallow graph neural networks [34, 32]. Compared to the GRAND model used in [10], we reduce the hidden dimension from 162 to 81 to fit the model into the GPU in our lab.

| Model | depth ($T$) | GRAND-l [10] | GRAND++-l (ours) | Improvement from GRAND++-l (ours) |
|---|---|---|---|---|
| | 1 | $68.50 \pm 0.76$ | $\mathbf{68.79 \pm 0.35}$ | 0.29 |
| | 4 | $69.53 \pm 0.21$ | $\mathbf{69.68 \pm 0.38}$ | 0.15 |
| | 6 | $69.46 \pm 0.43$ | $\mathbf{69.71 \pm 0.24}$ | 0.25 |
| | 8 | $69.44 \pm 0.30$ | $\mathbf{69.61 \pm 0.28}$ | 0.17 |
| OGBN-arXiv | 32 | $67.44 \pm 0.59$ | $\mathbf{69.41 \pm 0.53}$ | 1.97 |
| | 64 | $63.47 \pm 0.28$ | $\mathbf{68.05 \pm 0.73}$ | 4.58 |
| | 96 | $55.95 \pm 1.24$ | $\mathbf{67.26 \pm 0.61}$ | 11.31 |

Table 12: Classification accuracy of the linear GRAND and GRAND++ models trained with different depth on the OGBN-arXiv graph node classification task. Compared to the GRAND model used in [10], we reduce the hidden dimension from 162 to 81 to fit the model into the GPU in our lab. (Unit: %)

