# OpenReview forum: "GRAND++: Graph Neural Diffusion with A Source Term"
_ICLR.cc/2022/Conference — ICLR 2022 Poster_

### Official Review · Reviewer_durM · 2021-11-02

**Correctness:** 4
**Technical Novelty And Significance:** 3
**Empirical Novelty And Significance:** Not applicable
**Recommendation:** 8
**Confidence:** 3

**Main Review:**

Strengths:
1) Overall, I feel this is a well-organized paper and I enjoy reading it. This paper describes clearly the motivations of their model design. It also provides complete theoretical results and examples to explain the claimed issue (over-smoothing) and how the proposed method could solve it.
2) The proposed model is simple and neat. By simply adding the constant source term to the original diffusion equation and adjusting the initial conditions, this model is able to prevent the over-smoothing issue of diffusive GNN models. Moreover, this model could reuse the design and implementations of its source-term free version (GRAND).
3) The ablation study plots suggest that this model is able to effectively alleviate the important problem of over-smoothing issue of GNN models and benefits from adding more depth to the network.

Weakness:
1) The accuracy gains are not stable. The proposed method has worse classification accuracy than GNN and GCN significantly in the CoauthorCS dataset. The causes remain unexplained. Also, on CORA and CiteSeer the added source-term are worse than the baseline GRAND model.
2) This paper uses the classification with low-labeling rate as the prediction task to validate the effectiveness of preventing over-smoothing. However, over-smoothing could happen even without low-labelling rate. It is not clear whether the proposed method could practically solves the over-smoothing problems.

**Summary Of The Paper:**

This paper proposes a novel graph diffusion model with a source term in order to solve the over-smoothing issue of GRAND model and GNNs with diffusive property when the depth of the GNNs is large. This design also leads to a class of more accurate GNN based classification models when the labeling rate in the graph is low.

**Summary Of The Review:**

In general, this is a solid paper which proposes a novel idea to solve the over-smoothing problem the GNN models using a special form of graph diffusion equation. Solid theoretical and experimental results are provided in the paper.

---

> ### Author Response · Authors · 2021-11-15
> **Response to Reviewer durM**
>
> Thank you for your thoughtful review, endorsement, and valuable feedback. We are pleased that the reviewer enjoys reading the paper. Below we address your concerns.
>
> -----
>
> **Q1. The accuracy gains are not stable. The proposed method has worse classification accuracy than GNN and GCN significantly in the CoauthorCS dataset. The causes remain unexplained. Also, on CORA and CiteSeer the added source-term are worse than the baseline GRAND model.**
>
> **Reply:** The results of GRAND++ in Table 2 are conducted using the $T$ provided by the GRAND paper’s authors, which is not optimal for GRAND++. This is partially why GRAND++ does not outperform GRAND occasionally, particularly when the labeling rate is not low. However, when the labeling rate is low, e.g., one labeled node per class. GRAND++ tends to perform significantly better than GRAND. In the revised paper, we have reported the results of GRAND++ using a slightly fine-tuned $T$ based on a coarse search around the optimal value for GRAND. As you can see in Table 1, GRAND++ outperforms GRAND more. Moreover, we have added a t-test in Appendix D.5 to show the statistical significance of the accuracy gain as suggested by other reviewers.
>
> Another major advantage of GRAND++ over GRAND is alleviating over-smoothing. Due to the page limit, we placed the more detailed results in Table 5 in the appendix. As you can see from the results in Table 5, GRAND++ significantly outperforms GRAND and other baseline models, especially when $T$ is very large.
>
> The worse accuracy compared to GCN and GraphSage on CoauthorCS is worth exploring. Although we do not fully understand the cause yet, we suspect that the worse accuracy comes from some unknown issues in GRAND that are worth studying.
>
> -----
>
> **Q2. This paper uses the classification with low-labeling rate as the prediction task to validate the effectiveness of preventing over-smoothing. However, over-smoothing could happen even without low-labelling rate. It is not clear whether the proposed method could practically solves the over-smoothing problems.**
>
> **Reply:** Besides proving that the graph node features will not converge to a constant vector across nodes, we also numerically test the performance of GRAND++ with different labeling rates and different depths, see Figure 2 and Table 5 in the appendix. The results in Table 5 shows that when the networks are deep, GRAND++ significantly outperforms GRAND and other graph neural networks, including GCN, GAT, and GraphSage. Moreover, the accuracy gains of using GRAND++ over other models become more significant as the network depth increases.
>
> -----
>
> We look forward to and appreciate your further feedback.

---

### Official Review · Reviewer_qrMB · 2021-11-02

**Correctness:** 4
**Technical Novelty And Significance:** 3
**Empirical Novelty And Significance:** 2
**Recommendation:** 6
**Confidence:** 4

**Main Review:**

Strengths: The problem faced by the paper is interesting and timely and the proposed approach seems reasonable. The article is well written, the method is clearly described, and the overall quality is good. The authors also provide the source code to facilitate experimental replication.

Weakness:
1. This paper is more like an external version of GRAND. The author spends a lot of time on background knowledge and GRAND's work, which may undermine the contribution of this paper.
2. The assumption of Proposition 2 and 3 seems a little bit strong. We cannot guarantee how the graph data will look like, nor can we guarantee that the graph is connected.
3. I have some doubts about the 5.2 experiment, why use a setting where each class has only one label. In the above experiment, the authors used 20 labels per class, while in this experiment just 1 label was used. I would like to have some intermediate stage of experiments, like 5/10 labels per class.


**Summary Of The Paper:**

In this paper, the authors propose a graph deep learning method called GRAND++. This framework based on GRAND and can work with a limited number of labeled nodes. Experiments are also conducted to demonstrate the effectiveness of the method.

**Summary Of The Review:**

The overall quality of the paper is good, with a clear narrative. The problems addressed are also much needed. The methodology is straightforward and clear, and there is also theoretical guarantee. So I recommend it for acceptance.

---

> ### Author Response · Authors · 2021-11-15
> **Response to Reviewer qrMB**
>
> Thank you for your thoughtful review, endorsement, and valuable feedback. Below we address your concerns.
>
> -----
>
> **Q1. This paper is more like an external version of GRAND. The author spends a lot of time on background knowledge and GRAND's work, which may undermine the contribution of this paper.**
>
> **Reply:** Thanks for pointing this out. We have restructured the paper to avoid giving readers the impression of too much review of existing work. In particular, we have split the original “GRAND and its Random Walk Interpretation” section into two sections, i.e., “A Brief Review of GRAND” and “Random walk viewpoint of GRAND.”
>
> The section “A Brief Review of GRAND” is devoted to reviewing the formulation of GRAND and some detailed terminology. The section “Random walk viewpoint of GRAND”  presents a random walk viewpoint of GRAND and corresponding theoretical results. The random walk viewpoint of GRAND and corresponding theoretical results are new. They are crucial for analyzing potential over-smoothing of GRAND and serve as a motivation of GRAND++.
>
> -----
>
> **Q2. The assumption of Proposition 2 and 3 seems a little bit strong. We cannot guarantee how the graph data will look like, nor can we guarantee that the graph is connected.**
>
> **Reply:** The analysis can be extended to a graph with multiple connected components, in which case the node features converge to a vector that is constant on each connected component. Moreover, when the graphs are disconnected then each connected component behaves independently; so our analysis can be applied to each connected component separately.
>
> -----
>
> **Q3. I have some doubts about the 5.2 experiment, why use a setting where each class has only one label. In the above experiment, the authors used 20 labels per class, while in this experiment just 1 label was used. I would like to have some intermediate stage of experiments, like 5/10 labels per class.**
>
> **Reply:** We actually have compared GRAND++ with other models with different numbers of labeled nodes per class, including 1, 2, 5, 10, and 20. See Table 1 for the detailed results. In Figure 3, we only plotted the results for one labeled node per class.
>
> -----
>
> We look forward to and appreciate your further feedback.

---

> > ### Comment · Reviewer_qrMB · 2021-11-28
> > **Thanks for your reply**
> >
> > Thanks for the author's reply and these have answered my questions.The authors have also modified the paper accordingly and I am satisfied with the revised version.

---

> > > ### Author Response · Authors · 2021-11-28
> > > **Thanks for your responses**
> > >
> > > Thanks for your responses and we appreciate your endorsement.

---

### Official Review · Reviewer_p1wb · 2021-11-03

**Correctness:** 3
**Technical Novelty And Significance:** 3
**Empirical Novelty And Significance:** 2
**Recommendation:** 6
**Confidence:** 4

**Main Review:**

The paper addresses an important issue which affects neural ODEs and deep NN on graphs. In particular, concerning the GRAND differential equation, I think that adding a source term is an important idea essential to obtain a non-trivial long term behaviour of the dynamical system. This is clear when looking at linear dynamics (as done in the paper) as it is well known that the system converges to the dominant eigenmodes in that case. In fact, the addition of a source term is typical of most diffusion models on graphs, including standard Label Spreading/Label Propagation and their variants.

Weaknesses:
The paper is sometimes poorly written with several typos affecting the mathematics. I list below some of the major points that require improvement to my opinion:
- In the notation, the Hadamard product should be explicitly defined (or maybe just say that this is entrywise multiplication)
- Eq (1) uses notations that are not mentioned in the main paper such as $\nabla$ or $\mathrm{div}$ operators (they are in the supp material). I believe it would be very helpful to recall what this operators are in the main text.
- Right after eq (1): you consider the simplest case where $G$ is only dependent on the initial $X=X(0)$. However, this is not clear. This is a particular choice of $G$ that does not change in time, it is not the only one. Also, your choice seems to have no dependence on $X$ but rather on $W$ (the graph structure). So this seems to be the case of a $G$ that does not depend on $t$ NOR on $X$. In fact, soon afterwards at the beginning of $\S 3$ you consider another case where there is no dependence on $t$ but there is dependence on $X$ via the attention matrix $A(X)$. Also, what do you mean by "so we pick $G_{ij} = 1/d_i$? When do you pick this? Overall, I find this paragraph very confusing
- Why does the step size in Euler depend on $t$? It seems to me this is just a constant in your subsequent derivations
- Before eq (6) you quote eq (2) while I think you should refer to eq (1)
- You define GRAND-l after eq (7) but this definition is not clear to me. Right before you say that GRAND is based on the solution of (5) via ODE solvers. But then, when you define GRAND-l you refer to the attention matrix $A(X)$, which may not appear in (5). As you will use GRAND-l often in the rest of the paper, I believe this deserves a more clear definition here
- In the definition of the random walk (8), is $x^{(i)}=x^{(i)}(0)$? Also, this is quite unusual definition: typically the state space of a random walk on a graph are the nodes not on the their feature embedding. What is the advantage of defining the RW on the features?
- Again, the notation in Proposition 3 is not clear: what is $x^{(j)}$?
- Equation (1): what does "the source at feature vector of node $j$" mean?
- Starting from section 4 you move from $X(t)$ to $Z(t)$ in a way that is to my opinion inconsistent and not clear.
- In the centred equation after (10) there are several typos: I think $x$ should be $z$ or maybe you should use $x$ in (10) rather than $z$? $\tilde x$ was $\bar x$ in prop 3; I think your approximate dynamics should be $-x^{(i)}(t) + A\bar x$ rather than $-x^{(i)}(t) + \tilde x$, or could you clarify why this is not the case?
- Right after that equation you state: "for $i\in I$ we choose $c$ so that $z'(t)\approx 0$. This is not clear to me. What is $c$? What does $z'(t)\approx 0$ mean?
- The statement of Proposition 5 is quite sloppy. First, it is not clear what $z(k\delta_t)$ is? Is this the one from (14)? Second, mathematically, "it does not converge to a constant vector" makes not much sense? For example, if constant means that the limit does not depend on $t_0$, then this is probably not true. If it means that it has all constant entries, than still this might no be true as it would depend on the initial feature space. A much more precise statement to my opinion would say what the sequence converges to, rather than what it does not.
- Concerning this convergence questions, overall I do not really understand why you are considering the convergence of the sequence obtained via Euler integration, rather than the real dynamical system. Since you are anyway considering the simplest case of a linear dynamical system, why don't you consider the exact solution and study its behaviour? The fact that Euler converges to something might not mean that any numerical integrator would
- As stated above, in the experiments you mention GRAND-l and GRAND++-l but it is not clear to me what exactly are those



**Summary Of The Paper:**

The paper proposes a modification of the continuous-time graph neural network architecture GRAND where a source term is added to the differential equation in order to (a) avoid convergence of the dynamics towards constant vectors and thus mitigate the over-smoothing effect which affects the long time behaviour of the original dynamical system and graph neural networks with many layers in general and (b) increase classification performance under limited number of training points.

**Summary Of The Review:**

I like very much the idea of modifying the Laplacian diffusion equation div-grad used in GRAND adding a source term. The theoretical analysis proposed in the paper deals mostly with the simplest case of a linear diffusion operator, which is certainly useful to gain intuition. However, no attempt to transfer the results to more general nonlinear settings are discussed. Moreover, the mathematics contains many typos and a number of statements that are vague or not clear.

---

> ### Author Response · Authors · 2021-11-15
> **Response to Reviewer p1wb (1/3)**
>
> Thank you for your thoughtful review and valuable feedback. Below we address your concerns. We are pleased that the reviewer likes very much of the GRAND++ idea.
>
> -----
>
> **Q1. 1)  In the notation, the Hadamard product should be explicitly defined (or maybe just say that this is entrywise multiplication).  2) Eq (1) uses notations that are not mentioned in the main paper such as $\nabla$ or $\mathrm{div}$ operators (they are in the supp material). I believe it would be very helpful to recall what these operators are in the main text.**
>
> **Reply:** Thanks for your suggestion. As you suggested, we have included the explanation of the Hadamard product in the revised paper. Also, we have recalled the operator $\nabla$ or $\mathrm{div}$ right after Eq (1).
>
> -----
>
> **Q2. Right after eq (1): you consider the simplest case where $\mathbf{G}$ is only dependent on the initial ${\bf X}={\bf X}(0)$. However, this is not clear. This is a particular choice of $\mathbf{G}$ that does not change in time, it is not the only one. Also, your choice seems to have no dependence on ${\bf X}$ but rather on ${\bf W}$ (the graph structure). So this seems to be the case of a ${\bf G}$ that does not depend on $t$ NOR on ${\bf X}$. In fact, soon afterwards at the beginning of Sec. 3 you consider another case where there is no dependence on $t$ but there is dependence on ${\bf X}$ via the attention matrix ${\bf A}({\bf X})$. Also, what do you mean by "so we pick $G_{ij}=1/d_i$? When do you pick this?**
>
> **Reply:** The choice of ${\bf G}$ depends on the initial node features, and we followed the same setting of the GRAND paper. Let us clarify this below:
>
> First, in Sec. 2, we reviewed the diffusion equation on graphs, which can be generally formulated as Eq (1). The simplest case we discussed was when ${\bf G}$ only depends on the initial node features; in this case, with a particular choice of ${\bf G}$ we can reformulate the graph diffusion as Eq (2). However, we agree with the reviewer that we have made a further choice by picking $G_{ij}=1/d_i$ and we have made this clear in the revised paper. We can directly learn the matrix ${\bf A}$, which again only depends on the initial node features ${\bf X}$, i.e. ${\bf A}={\bf A}({\bf X})$. In this case ${\bf G}$ does not depend on ${\bf X}$ or $t$ and does depend on ${\bf W}$ as the referee observes. However, we note that in Eq (1) we pointwise multiply ${\bf G}$ by $\nabla {\bf X}$ and so the PDE in Eq (1) depends linearly on ${\bf X}$.
>
> Second, we use the self-attention mechanism in Eq (7) to learn the matrix ${\bf A}$. Note that Eq (7) only depends on the initial node features. We have made this point more clearly in the revised paper by stressing that ${\bf A}$ depends on ${\bf X}$ right after Eq (2).
>
> Picking $G_{ij}=1/d_i$ is to construct the random walk Laplacian given the matrix ${\bf W}$. But in GRAND and GRAND++, we do not need to deal with this explicitly. We directly model the matrix ${\bf A}={\bf D}^{-1}{\bf W}$ by using the self-attention mechanism, which guarantees the sum of each row is one.
>
> -----
>
> **Q3. Why does the step size in Euler depend on $t$? It seems to me this is just a constant in your subsequent derivations.**
>
> **Reply:** Here we used $\delta_t$ is to stress is the step size for discretizing the time derivative.
>
> -----
>
> **Q4. Before eq (6) you quote eq (2) while I think you should refer to eq (1).**
>
> **Reply:** Both are fine but referring to (1) is better. Eq (2) is for the case when ${\bf G}$ depends only on the initial node features, which is equivalent to Eq (6). Eq (6) only explicitly points out what is $-\mathbf{L}$ in Eq (2). We have referred to eq (1) in the revision.
>
> **Q5. You define GRAND-l after eq (7) but this definition is not clear to me. Right before you say that GRAND is based on the solution of (5) via ODE solvers. But then, when you define GRAND-l you refer to the attention matrix ${\bf A}({\bf X})$, which may not appear in (5). As you will use GRAND-l often in the rest of the paper, I believe this deserves a more clear definition here.**
>
> **Reply:** Here we adopt the notation from the original GRAND paper. GRAND-l is the model in Eqs (5), (6), (7); that is, the diffusivity is modeled by the self-attention mechanism in Eq (7), which only depends on the initial graph node features.
>
> We have added more discussion to clarify GRAND and GRAND-l. In a nutshell, GRAND-l is a special case of GRAND, when diffusivity depends only on the initial graph node features.

---

> > ### Author Response · Authors · 2021-11-15
> > **Response to Reviewer p1wb (2/3)**
> >
> > **Q6. In the definition of the random walk (8), is ${\bf x}(i)={\bf x}^{(i)}(0)$? Also, this is quite unusual definition: typically the state space of a random walk on a graph are the nodes not on the their feature embedding. What is the advantage of defining the RW on the features?**
> >
> > **Reply:** Yes, we have emphasized that the random walk is defined over ${\bf x}^{(i)}(0)$ in the revision. In Section 4 the results hold for any choice of ${\bf x}^{(i)}(0)$ (it only matters that the random walk is defined on the graph with the same nodes as used to initialise the differential equation).
> >
> > We define such a random walk to describe and analyze the diffusion equation on graphs, and whilst it may not be common to see a random walk on features using a diffusion equation is not new. Note that the diffusion equation we considered is a diffusion process of graph node features.
> >
> > -----
> >
> > **Q7. Again, the notation in Proposition 3 is not clear: what is ${\bf x}^{(j)}$?**
> >
> > **Reply:** ${\bf x}^{(j)}$ is the initial node feature of the $j$-th node. See the beginning of Section 2, where the graph we considered is $({\bf X},{\bf W})$ and the node features are ${\bf X}=([{\bf x}^{(1)}]^\top,\cdots,[{\bf x}^{(n)}]^\top)^\top$.
> >
> > -----
> >
> > **Q8. Equation (10): what does "the source at feature vector of node $j$" mean?**
> >
> > **Reply:** It means we put a source term at the $j$-th graph node, which has the initial node feature ${\bf x}^{(j)}$. Note that here we considered the diffusion on the graph, so we can place a source term at any graph node.
> >
> > -----
> >
> > **Q9. Starting from section 4 you move from ${\bf X}(t)$ to ${\bf Z}(t)$ in a way that is to my opinion inconsistent and not clear.**
> >
> > **Reply:** Here we use ${\bf Z}(t)$ to describe GRAND++ to distinguish it from GRAND, which we used ${\bf X}(t)$. To avoid confusion, in the revision, we have pointed out that we use ${\bf X}(t)$ and ${\bf Z}(t)$ to describe GRAND and GRAND++ dynamics, respectively.
> >
> > -----
> >
> > **Q10. In the centred equation after (10) there are several typos: I think ${\bf x}$ should be ${\bf z}$ or maybe you should use ${\bf x}$  in (10) rather than ${\bf z}$? $\tilde{\bf x}$ was $\bar{\bf x}$ in prop 3; I think your approximate dynamics should be $-{\bf x}^{(i)}(t)+A\bar{\bf x}$ rather than $-{\bf x}^{(i)}(t)+\tilde{\bf x}$, or could you clarify why this is not the case?**
> >
> > **Reply:** We used ${\bf x}$ and ${\bf z}$ to describe the dynamics of GRAND and GRAND++, respectively. The equation after (10) is used to characterize the bias that arises from the GRAND, which is the reason we used ${\bf x}$ rather than ${\bf z}$. In the revised paper, we have clarified why we use ${\bf z}$ for GRAND++-related models.
> >
> > There was a typo in eq. (8) and Proposition 3 where the random walk should have been defined on the graph with nodes ${\bf x}^{(i)}(0)$ for $i=1,\cdots,n$, rather than ${\bf x}^{(i)}$ for $i=1,\cdots,n$. This means the result of Proposition 3 is that the long time limit is to $\tilde{{\bf x}}$. With this correction the approximate dynamics given later are correct. Also, $\frac{1}{d_i}\sum_{j=1}^nW_{ij}=1$ implies that $\frac{1}{d_i}\sum_{j=1}^nW_{ij}\tilde{{\bf x}}=\tilde{{\bf x}}$
> >
> >
> >
> > -----
> >
> > **Q11. Right after that equation you state: "for $i\in I$ we choose $c$ so that ${\bf z}’(t)\approx 0$. This is not clear to me. What is $c$? What does ${\bf z}’(t)\approx 0$ mean?**
> >
> > **Reply:** It is a typo. It should be $C_i$.
> >
> > -----
> >
> > **Q12. The statement of Proposition 5 is quite sloppy. First, it is not clear what ${\bf z}(k\delta_t)$ is? Is this the one from (14)? Second, mathematically, "it does not converge to a constant vector" makes not much sense? For example, if constant means that the limit does not depend on $t_0$, then this is probably not true. If it means that it has all constant entries, than still this might no be true as it would depend on the initial feature space. A much more precise statement to my opinion would say what the sequence converges to, rather than what it does not.**
> >
> > **Reply:** ${\bf z}^{(i)}(k\delta_t)$ is the same as that in Eq (14), and we have added some descriptions of it in the revision. The statement “does not converge to a constant vector, for all $i$” was poorly phrased. We meant that the node features will not become the same across graph nodes under the GRAND++ dynamics (a better statement is that ${\bf z}^{(i)}(k\delta_t)$ does not converge to a constant as a function of $i$). We have also clarified this in the revision.
> >
> > We provided the limit of ${\bf z}^{(i)}(k\delta_t)$ when $k\rightarrow \infty$ in Proposition 4 and the interpretation of this limit is given in Remark 1.

---

> > > ### Author Response · Authors · 2021-11-15
> > > **Response to Reviewer p1wb (3/3)**
> > >
> > > **Q13. Concerning this convergence questions, overall I do not really understand why you are considering the convergence of the sequence obtained via Euler integration, rather than the real dynamical system. Since you are anyway considering the simplest case of a linear dynamical system, why don't you consider the exact solution and study its behaviour? The fact that Euler converges to something might not mean that any numerical integrator would.**
> > >
> > > **Reply:** We performed a random walk analysis of the behavior of the GRAND++ dynamics. Random walk analysis of the behavior of diffusion equations is relatively standard and popular. Another reason we used the random walk analysis is that we are considering diffusion equations on graphs; the standard analysis of the Euclidean space does not apply.
> > >
> > > As long as the numerical integrator is consistent and stable, i.e., convergent, the behavior should be the same as our theoretical results.
> > >
> > > One could, of course, study the exact solution to the dynamical system via analysing a continuous-time random walk (i.e. a Brownian motion). To analyse the Brownian motion one would likely have to perform a discretization in time which would be equivalent to studying the Euler discretisation but with the notational cost of working with continuous-time stochastic processes, which we feel would obscure the main ideas.
> > >
> > > -----
> > >
> > > **Q14. As stated above, in the experiments you mention GRAND-l and GRAND++-l but it is not clear to me what exactly are those.**
> > >
> > > **Reply:** GRAND-l is the model in Eqs (5), (6), (7); that is, the diffusivity is modeled by the self-attention mechanism in Eq (7), which only depends on the initial graph node features.
> > >
> > > GRAND++-l is a modification of GRAND-l by adding a source term to the governing diffusion equation and changing the initial condition of GRAND-l. The governing equation in GRAND++-l is given in Eq (13).
> > >
> > > -----
> > >
> > > **Q15. I like very much the idea of modifying the Laplacian diffusion equation div-grad used in GRAND adding a source term. The theoretical analysis proposed in the paper deals mostly with the simplest case of a linear diffusion operator, which is certainly useful to gain intuition. However, no attempt to transfer the results to more general nonlinear settings are discussed.**
> > >
> > > **Reply:** The general nonlinear setting is very difficult to analyze. As far as we know, analyzing the behavior of general nonlinear diffusion equations is also wide open in the theoretical PDE analysis community. The random walk framework we developed cannot be easily adapted to the general nonlinear settings. However, we conjecture that adding a source term to the general nonlinear diffusion equation also alleviates over-smoothing for general nonlinear settings. At least in the simple linear setting we are able to motivate our choice of architecture, but of course it would be even better to be able to extend these results in the nonlinear setting. Since, as far as we are aware, there are no tools for such an analysis we instead provide a few numerical comparisons of GRAND and GRAND++ in nonlinear settings in Table 2, which shows that our designed source term also improves GRAND in nonlinear settings.
> > >
> > > From the practical viewpoint, both the GRAND authors and we noticed that the linear graph neural diffusion usually outperforms the nonlinear models.
> > >
> > > -----
> > >
> > > We look forward to and appreciate your further feedback.

---

> > > > ### Comment · Reviewer_p1wb · 2021-11-21
> > > > **Thank you for your responses.**
> > > >
> > > > Thanks for your timely and detailed responses.
> > > >
> > > > About Q13, it still not clear to me why can't you solve a linear ODE $\dot x = M(x(0))x$ explicitly. Can you explain what is the added difficulty given by the fact that $M(x(0))$ has to do with a graph?
> > > >
> > > > About Q15, I agree that studying nonlinear dynamics is way more challenging than the linear case. However,
> > > > 1) there are ways to analyse the long-term behaviour of nonlinear dynamical systems via e.g. index/bifurcation theory
> > > > 2) if you only study the linear case and you conjecture properties transfer to the nonlinear setting, you should explain why you make this conjecture. I guess you would need to look at the gradient of the nonlinear operator, and this would not hold for all nonlinearities.
> > > >
> > > > In any case, I am OK overall with the way you addressed my concerns ans those of the other reviewers and I will adjust my score accordingly

---

> > > > > ### Author Response · Authors · 2021-11-22
> > > > > **Response to Reviewer p1wb**
> > > > >
> > > > > Thanks for your further feedback and we appreciate your endorsement.
> > > > >
> > > > > -----
> > > > >
> > > > >
> > > > > ***Q1. About Q13, it still not clear to me why can't you solve a linear ODE $\dot x = M(x(0))x$ explicitly. Can you explain what is the added difficulty given by the fact that $M(x(0))$ has to do with a graph?***
> > > > >
> > > > > ***Answer:*** Indeed, we can solve the linear ODE $\dot x = M(x(0))x$, where $M(x(0))$ is semi-negative definite with only one zero eigenvalue. The long-time behaviour of $x(t)$ is dominated by the leading eigenmode of $M(x(0))$, and from this perspective we can also show that GRAND suffers from over-smoothing.
> > > > >
> > > > > The random walk framework enables us to unify the analysis of GRAND and GRAND++. It is certainly interesting to improve the GRAND model from the nonhomogeneous ODE viewpoint.
> > > > >
> > > > > ----
> > > > >
> > > > > ***Q2. About Q15, I agree that studying nonlinear dynamics is way more challenging than the linear case. However, 1. there are ways to analyse the long-term behaviour of nonlinear dynamical systems via e.g. index/bifurcation theory. 2. if you only study the linear case and you conjecture properties transfer to the nonlinear setting, you should explain why you make this conjecture. I guess you would need to look at the gradient of the nonlinear operator, and this would not hold for all nonlinearities.***
> > > > >
> > > > >
> > > > > ***Answer:*** Thank you for pointing these out. These are certainly very interesting and worth further study. Based on our experiments and the results from the original GRAND paper, both nonlinear GRAND and GRAND++ usually perform inferior to the linear counterparts. We believe much more effort is needed to improve nonlinear GRAND or GRAND++.
> > > > >
> > > > > -----
> > > > >
> > > > > We thank you again for your valuable feedback and endorsement.

---

### Official Review · Reviewer_zJ38 · 2021-11-07

**Correctness:** 3
**Technical Novelty And Significance:** 2
**Empirical Novelty And Significance:** 2
**Recommendation:** 6
**Confidence:** 3

**Main Review:**

Pros:

The authors provide comprehensive background knowledge for the different parts of their algorithem.

They theoretically show that the source term helps alleviate the over-smoothing problem.

The experiment regarding limited label data is interesting.

Cons and concerns:

One of my main concerns is the empirical results reported in the paper. Specifically, I would be grateful if the authors could elaborate on why there is a mismatch between the results reported for the "GRAND" method (when for each label, 20 samples were used) in the original paper and the numbers reported in the current submission. For instance, on CORA dataset, the GRAND paper reported 83.6 ± 1.0 while the submission reported GRAND test accuracy as 82.86 ± 2.39. This is important because the test accuracy of GRAN++ is 82.95 ± 1.37.




To actually show that GRAND++ is empirically better than other methods, the authors need to do a t-test. That is because std. of the reported test accuracy is too high (16 percent for CORA) when the number of observed samples for each label is less than 20. Note, this is different from having a small improvement; the t-test has to be done to show whether there is actually an improvement or not.



Lack of experiments on datasets that needs larger depth NN. Authors motivate their method by mentioning that they avoid over-smoothing; however, there must be an empirical need to develop such methods.  I suggest authors to experiment on a dataset that needs large depth to show the importance of their contribution.




**Summary Of The Paper:**

This paper studies neural-based diffusion models for graph data. It considers deep learning on graphs as a continuous diffusion process and treats GNNs as discretizations of an underlying PDE. The authors build upon an existing work (GRAND) by adding a source term to the objective function. The source term is designed to alleviate the over-smoothing problem of GNN, a well-known phenomenon that deeper GNN learns similar representation for each node. They theoretically show the effectiveness of the source term in mitigation over-smoothing problem. Furthermore, they empirically assess their method's effectiveness in learning with limited labeled data and using deep architectures on several benchmark datasets.

**Summary Of The Review:**

Please look at the comments provided above.

---

> ### Author Response · Authors · 2021-11-15
> **Response to Reviewer zJ38**
>
> Thank you for your thoughtful review and valuable feedback. Below we address your concerns.
>
> -----
>
> **Q1. One of my main concerns is the empirical results reported in the paper. Specifically, I would be grateful if the authors could elaborate on why there is a mismatch between the results reported for the "GRAND" method (when for each label, 20 samples were used) in the original paper and the numbers reported in the current submission. For instance, on CORA dataset, the GRAND paper reported 83.6 $\pm$ 1.0 while the submission reported GRAND test accuracy as 82.86 $\pm$ 2.39. This is important because the test accuracy of GRAN++ is 82.95 $\pm$ 1.37.**
>
> **Reply:** There is a trick that GRAND used to obtain better results reported in the GRAND paper for some of the datasets.  They added node features of the output of the first layer to the ODE equation while detaching this additional term (treating it as a parameter independent constant when computing gradient). (See lines 64 in their released code at “https://github.com/twitter-research/graph-neural-pde/blob/0a5bcc7121762e5578ce95a1dae19f1464f88950/src/GNN_early.py .”) This trick is task-specific: it can improve the performance on some datasets while harming most others. Because we can’t find references and reasons on when and why this trick works, we did not include this trick in the experiments for a fair comparison.
>
> We can also integrate the above trick into GRAND++, and in that case, GRAND++ has an accuracy of 84.35 $\pm$ 1.16 for CORA.
>
> -----
>
> **Q2. To actually show that GRAND++ is empirically better than other methods, the authors need to do a t-test. That is because std. of the reported test accuracy is too high (16 percent for CORA) when the number of observed samples for each label is less than 20. Note, this is different from having a small improvement; the t-test has to be done to show whether there is actually an improvement or not.**
>
> **Reply:** Thanks for your suggestion. In the revision, we have added a t-test to show the statistical significance of the accuracy gain of GRAND++. See Appendix D.5 for details.
>
> -----
>
> **Q3. Lack of experiments on datasets that needs larger depth NN. Authors motivate their method by mentioning that they avoid over-smoothing; however, there must be an empirical need to develop such methods. I suggest authors to experiment on a dataset that needs large depth to show the importance of their contribution.**
>
> **Reply:** We used the same experiments used in the GRAND paper, which also aims to mitigate the over-smoothing issue. We note that these tasks are widely used to evaluate the efficacy of overcoming over-smoothing.
>
> Moreover, as you suggested, we have added more results on ogbn-arxiv classification in the revised paper, see Appendix D.6. The ogbn-arxiv classification task is a standard benchmark for evaluating the efficacy of deep graph neural networks (Li and et al., Training Graph Neural Networks with 1000 Layers, ICML, 2021). Again, the results show that GRAND++ outperforms GRAND.
>
> -----
>
> We look forward to and appreciate your further feedback.

---

> > ### Comment · Reviewer_zJ38 · 2021-11-28
> > **Response to authors**
> >
> > I thank the authors for providing a detailed explanation and additional experiments to address my concerns. After reading the author's response and the other reviewers' comments, I decided to increase my score from 5 to 6.

---

> > > ### Author Response · Authors · 2021-11-28
> > > **Thank you**
> > >
> > > Thanks for your responses and we appreciate your endorsement.

---

### Official Review · Reviewer_y1z1 · 2021-11-08

**Correctness:** 3
**Technical Novelty And Significance:** 2
**Empirical Novelty And Significance:** 2
**Recommendation:** 6
**Confidence:** 3

**Main Review:**




Pros:
- The problem is, in my opinion, both interesting and relevant for the research community
- The model seems to perform particularly well when there are very, very few labeled data points

Cons:
- A few parts are introduced without much context or references in the text, e.g. Laplace and Poisson learning are cited as related work but the connection to the present one is not clear enough IMO (except for the problem of inference inconsistency at low labeling rate)

- Formulation is not always clear, e.g.:
  - in Eq (3), $\tilde L$ is introduced as a shorthand for $(I-\delta_{t}L)$. Does this matrix have any particular interpretation?
  - in page 4, between (6) and (7), $h$ is used both as an index for layers and as the max value (something like $\sum_{l=1..h} A^{l}(X)$ or even just using $l$ would make it more readable)
  - the random walk definition in (8) did not convince me (perhaps because it was introduced with no further explanation). How is the step size $\delta_t$ defined? To me it should not be possible to use it as in $1 - \delta_{t}$ to define a probability value, as there was no definition of $T$ thus it can take any value greater than zero.
  - in Eq (10.5), $\pi_{j}=: \tilde{x}$ shouldn't this be $\bar{x}$ instead?

- experimental results are, in my opinion, not convincing:
  - related to the claim of performing better with fewer labeled data, while there are some datasets in which the method actually does well there also many cases (CoauthorCS, most of Computer, and Photo) in which it is worse or comparable to simpler approaches (learning the reasons why this happens, though, would be interesting per se)
  - still related to numbers, the noise of low-labels dataset is such (and the variance is so high) that even results out of 2000 experiments are not statistically very significant: some of them (e.g. CORA-20, PubMed-2, Photo-20) have a p-value > 0.1.
  - it is not clear why depth should be such a useful treat when both GRAND and GRAND++ have deteriorating performances when it grows (e.g. in Figure 2, 1-3 from left). In these cases GRAND seems to perform better or comparably most of the times (with the advantage of being simpler). Plots with small amount of labeled samples, instead, look more convincing in my opinion (i.e. they clearly show a use case in which GRAND++ would be preferable)


**Summary Of The Paper:**

The paper introduces GRAND++, GRAph Neural Diffusion with a source term. As its predecessor GRAND, this method is focused on the development of a new continuous-depth GNN to tackle the over-smoothing and the bottleneck issues which are typical of deep GNNs.
The authors introduce a random-walk interpretation of GRAND, which shows that when the network is very deep it is still prone to over-smoothing. To address this problem they add a source term to the problem formulation, which is shown to provide better results in very deep models and in classification tasks which are characterised by a low amount of labeled data.

**Summary Of The Review:**

I think the paper tackles an interesting problem and provides evidence that, in some extreme cases such as incredibly low amount of labels per class, it can outperform other approaches.

However, of its two main claims (avoiding over-smoothing and working well with few labels), I think the former is a bit harder to defend (GRAND++ is always better than GRAND when depth is very large but not always otherwise, in addition it is does not always exhibit the same behavior and it not clear why that happens). For the low-label case, I think there are more chances to convince about the improvement brought by the model but I think it would be important to better understand the reasons for it. Intuitively, a strong bias is introduced by the insertion of the source term that definitely helps by providing useful signal for the classification. This should be, in my opinion, discussed more in detail and verified with more examples.

For the above reasons, I consider the paper still not mature enough for publication but I think a more in-depth analysis could bring valuable results.

----

The authors' replies addressed my concerns and after reading their answers to other reviewers and the updates to their manuscript I decided to increase my rating.

---

> ### Author Response · Authors · 2021-11-15
> **Response to Reviewer y1z1 (1/3)**
>
> Thank you for your thoughtful review and valuable feedback. Below we address your concerns.
>
> -----
>
> **Q1. A few parts are introduced without much context or references in the text, e.g. Laplace and Poisson learning are cited as related work but the connection to the present one is not clear enough IMO (except for the problem of inference inconsistency at low labeling rate).**
>
> **Reply:** Both Laplace learning and Poisson learning use node features to construct a graph and then perform label propagation by solving a Laplace or Poisson equation. However, graph neural networks first learn node representation by propagating initial node features through the graph neural network layers. Then graph neural networks perform prediction by output activation on the learned representation.
>
> GRAND and GRAND++ both belong to graph neural networks. Poisson learning is related to GRAND++ in the sense that both models add a source term during the propagation. Poisson learning adds Green’s function--a particular source term--to the node labels, but GRAND++ adds the source term to the node features. The source terms considered in the two models are fundamentally different. Also, GRAND++ requires a special design on the initial condition for the diffusion equation.
>
> We have added a detailed discussion of Laplace and Poisson learning. Also, we have stressed the relation and difference of existing work to our work.
>
> -----
>
> **Q2. In Eq (3), $\tilde{\bf L}$ is introduced as a shorthand for $({\bf I}-\delta_t {\bf L})$. Does this matrix have any particular interpretation?**
>
> **Reply:** $\tilde{\bf L}$ is a particular type of low-pass filter. We are not aware of any specific interpretation but it can be seen to be a weighted Laplacian, see reference [52] in our paper,  i.e., “Z. Shi, et al., Weighted Nonlocal Laplacian on Interpolation from Sparse Data, Journal of Scientific Computing, 2017.”
>
> -----
>
> **Q3. In page 4, between  (6) and (7), $h$ is used both as an index for layers and as the max value (something like $\sum_{l=1..h}A^l({\bf X})$ or even just using $l$ would make it more readable).**
>
> **Reply:** Thank you for your suggestion, and we have changed the notation as you suggested.
>
> -----
>
> **Q4. The random walk definition in (8) did not convince me (perhaps because it was introduced with no further explanation). How is the step size $\delta_t$ defined? To me it should not be possible to use it as in $1-\delta_t$ to define a probability value, as there was no definition of $T$ thus it can take any value greater than zero.**
>
> **Reply:** $\delta_t\in [0,1]$ is a small positive number, the same as the step size used in Eq (3). Here we used subscript $t$ to stress it is the step size for discretizing the time derivative. Since $\delta_t$ is a positive small number, $1-\delta_t>0$ and $1-\delta_t+\sum_{l\neq j} \frac{\delta_tW_{jl}}{d_j}=1$. Here, note that $W_{jl}\geq 0$, $W_{jj}=0$, and $d_j=\sum_{l=1}^nW_{jl}$. When $\delta_t=1$ this expression is well known (see for example Zhu, Ghahramani and Lafferty, 2003). We include the proof of our proposition in the appendix. We have clarified that $\delta_t\in [0,1]$ so that (8) does indeed define a probability distribution.
>
> -----
>
> **Q5. In Eq (10.5), $\pi_j =: \tilde{\bf x}$ shouldn’t this be $\bar{\bf x}$ instead?**
>
> **Reply:** We’ve amended our notation slightly in order to avoid any confusion between $\bar{\bf x}$ and $\tilde{\bf x}$. Previously, we denoted $\sum_{j=1}^n{\bf x}^{(j)}(0)\pi_j$ as $\tilde{\bf x}$ and $\sum_{j=1}^n{\bf x}^{(j)}\pi_j$ as $\bar{\bf x}$, but for the choice ${\bf x}^{(j)}(0)={\bf x}^{(j)}$ one has $\bar{\bf x} = \tilde{\bf x}$ then we felt it unnecessary and perhaps confusing to have both. We now just keep $\tilde{\bf x}=\sum_{j=1}^n{\bf x}^{(j)}(0)\pi_j$. However, as we also make use of the unweighted average of feature vectors over the subset $\mathcal{I}$ then we introduce the notation $\hat{\bf x}=\frac{1}{|\mathcal{I}|} \sum_{i\in\mathcal{I}} {\bf x}^{(i)}$.

---

> > ### Author Response · Authors · 2021-11-15
> > **Response to Reviewer y1z1 (2/3)**
> >
> > **Q6. Related to the claim of performing better with fewer labeled data, while there are some datasets in which the method actually does well there also many cases (CoauthorCS, most of Computer, and Photo) in which it is worse or comparable to simpler approaches (learning the reasons why this happens, though, would be interesting per see).**
> >
> > **Reply:** For the results in Table 1, i.e., comparing different models with different labeling rates, we used the $T$ that was fine-tuned for GRAND, which makes GRAND outperforms GRAND++ occasionally. In the revision, we have employed a $T$ that was slightly fine-tuned for GRAND++ based on the optimal value for GRAND. As you can see from the results in Table 1 of our revision, GRAND++ outperforms GRAND more. We believe that with the same effort of fine-tuning for $T$ as GRAND did, GRAND++ can do even better.
> >
> > In a few cases, GCN and GAT are more accurate than GRAND and GRAND++. Although we don’t fully understand the cause yet, we suspect that the worse accuracy comes from some unknown issues in GRAND that are worth studying.
> >
> > One possible way to further improve GRAND and GRAND++ is to leverage some recently developed neural ODE algorithms. We note that there have been many recent developments of neural ODEs to improve their classification accuracy; some of those techniques can be used to improve the performance of GRAND and GRAND++. For instance, stacked neural ODEs (S. Massaroli, Dissecting Neural ODEs, NeurIPS, 2020). We believe the continuous-depth graph neural networks have much more to explore, and they should be possible to beat GCN or GraphSage on the CoauthorCS dataset.
> >
> > -----
> >
> > **Q7. Still related to numbers, the noise of low-labels dataset is such (and the variance is so high) that even results out of 2000 experiments are not statistically very significant: some of them (e.g. CORA-20, PubMed-2, Photo-20) have a p-value > 0.1.**
> >
> > **Reply:** For each task with each given labeling rate, the 2000 experiments are from 100 random splits with 20 random seeds for each split of training, test, and validation dataset. The classification accuracy varies significantly across different splits, which increases the variance and makes the accuracy improvement look not statistically significant. In the revision, for experiments that don't pass unpaired t-test with existing data due to large variance, we conducted more runs and added paired t-test on the accuracy improvement in Appendix D.5, which shows the accuracy gain of GRAND++ over GRAND is statistically significant at a p-value $<$ 0.05 on low labeled datasets.
> >
> > -----
> >
> > **Q8. It is not clear why depth should be such a useful treat when both GRAND and GRAND++ have deteriorating performances when it grows (e.g. in Figure 2, 1-3 from left). In these cases GRAND seems to perform better or comparably most of the times (with the advantage of being simpler).**
> >
> > **Reply:** Increasing the depth of GRAND or GRAND++ sometimes can improve the classification accuracy. In particular, for training GRAND++ with low-labeling rates, based on our testing, deeper GRAND++ performs better on the benchmark tasks when the labeling rates are low. Moreover, as you can see from the results in Table 5 in the appendix, both GRAND and GRAND++ achieve optimal classification accuracy with moderate depths. The capability of building deeper graph neural networks provides more options for different applications. Also, being able to build a model that works well for even very large $T$ will enable us to explore other neural ODE techniques to enrich the expressivity of continuous-depth neural networks. One particular idea is to integrate the idea of stacked neural ODEs (S. Massaroli, Dissecting Neural ODEs, NeurIPS, 2020) to enhance the capacity of continuous-depth GNNs. We have also added more results on using GRAND and GRAND++ with different depths for the ogbn-arxiv classification to confirm the benefits of the deep models; ogbn-arxiv is a standard benchmark task for evaluating deep graph neural networks (Li and et al., Training Graph Neural Networks with 1000 Layers, ICML, 2021).
> >
> > Besides graph node classification, we noticed that graph neural networks have also been used for learning complex dynamical systems, see, e.g., “Pfaff et al. Learning mesh-based simulation with graph networks, ICLR 2021”, and “Li et al. Learning particle dynamics for manipulating rigid bodies, deformable objects, and fluids, ICLR, 2019”. We believe being able to build continuous-depth graph neural networks that are suitable for learning with very large $T$ will be very useful for learning the long-time behavior of complex physical systems. Also, continuous-depth graph neural networks are better for learning many physical systems than discrete graph neural networks considering the continuous profiling of most physical systems.

---

> > > ### Author Response · Authors · 2021-11-15
> > > **Response to Reviewer y1z1 (3/3)**
> > >
> > > **Q9. Plots with small amounts of labeled samples, instead, look more convincing in my opinion (i.e. they clearly show a use case in which GRAND++ would be preferable).**
> > >
> > > **Reply:** In Figure 3, we have contrasted the performance of GRAND and GRAND++ in the low labeling rate cases. We have stressed the advantage of GRAND++ in low labeling rates in the revision.
> > >
> > > -----
> > >
> > > **Q10. However, of its two main claims (avoiding over-smoothing and working well with few labels), I think the former is a bit harder to defend (GRAND++ is always better than GRAND when depth is very large but not always otherwise, in addition it is does not always exhibit the same behavior and it not clear why that happens).**
> > >
> > > **Reply:** Regarding alleviating over-smoothing, GRAND++ enjoys both theoretical and practical advantages over GRAND. Theoretically, we showed that the node features under the GRAND dynamics would converge to a constant vector across different nodes, but GRAND++ will not; see Proposition 2 for GRAND and Propositions 4 and 5 for GRAND++.
> > >
> > > Empirically, we compared the performance of GRAND++, GRAND, and several other benchmark GNNs with different depths in Figure 2 and Table 5 in the appendix, showing that GRAND++ consistently outperforms GRAND and other models when the model is deep. We have highlighted the corresponding results in the revision.
> > >
> > > Perhaps there was confusion with the results in Table 1, where we used a default depth for GRAND and GRAND++ and tested their performances when they were trained with different labeling rates.
> > >
> > > -----
> > >
> > > **Q11. For the low-label case, I think there are more chances to convince about the improvement brought by the model but I think it would be important to better understand the reasons for it. Intuitively, a strong bias is introduced by the insertion of the source term that definitely helps by providing useful signal for the classification. This should be, in my opinion, discussed more in detail and verified with more examples.**
> > >
> > > **Reply:** Thanks for your suggestion; we have added more discussion about the source term in GRAND++ in the revision. In particular, the special source term guarantees the existence of stationary distribution, and the node features of the stationary distribution are not a constant vector over different nodes, which benefits overcoming over-smoothing. Also, in Remark 2, we explain why GRAND++ works even when the labeling rate is very low. We have highlighted the corresponding part in the revision.
> > >
> > > -----
> > >
> > > We look forward to and appreciate your further feedback.

---

### Author Response · Authors · 2021-11-15
**General Response and Summary of the Revision**

Dear AC and reviewers,

Thanks for your thoughtful reviews and valuable comments, which have helped us improve the paper significantly. We are encouraged by the endorsements that: 1) Our idea is novel (durM) and clear (qrMB) with theoretical guarantees (durM, qrMB), and we appreciate that reviewer p1wb likes much of the idea. 2) The experiment on limited data is interesting (zJ38) and performs well (y1z1). We have updated our submission based on the reviewers' feedback, and we have highlighted our revision in blue.

One of the common comments is that the performance gain of GRAND++ over GRAND is sometimes not very significant. We first address this comment here. GRAND++ has two advantages over GRAND and other graph neural networks:

- 1. GRAND++ can help overcome the over-smoothing issue since the node features will not converge to a constant vector across the graph nodes. Numerically, we verified this fact with networks of different depths, as you can see in **Figure 2 and Table 5 in the appendix**. The performance gain of GRAND++ over other models is quite significant, especially when the network is very deep.

- 2. GRAND++ is more effective in learning with low-labeling rates, and the reason is given in **Remark 2** of our paper. The numerical results are available in **Figure 3 and Table 1**. Here, for GRAND++ we used the fine-tuned $T$ for GRAND, which may not be optimal for GRAND++. In the revision, we have reported the result of GRAND++ using a slightly fine-tuned $T$ based on a small search around the optimal value for GRAND, see Table 1. As you can see that the performance gain is quite significant for learning with low-labeling rates.

Also, there might be confusion about the purpose of the results in each figure and table. We stress that the results in **Figure 2 and Table 5 in the appendix** are used to show GRAND++ is effective in overcoming over-smoothing and suited for learning with very deep architectures. While the results in **Figure 3 and Table 1** show that GRAND++ is more effective in learning with low-labeling rates. We have added some explanations in the caption of these figures and tables to ease understanding of these results.

-----

Incorporating the comments and suggestions from all reviewers, besides fixing typos and notations, we have made the following changes in the revised paper.

- 1. As Reviewer qrMB pointed out, the original section “GRAND and its Random Walk Interpretation” gives the reader an impression of spending a lot of time on background knowledge and GRAND’s work, which may undermine the contribution of our paper. We have split the original “GRAND and its Random Walk Interpretation” section into two sections, i.e., “A Brief Review of GRAND” and “Random walk viewpoint of GRAND.” The section “GRAND and its Random Walk Interpretation” is devoted to reviewing the formulation of GRAND and some detailed terminology. The section “Random walk viewpoint of GRAND” is our new result, which presents a random walk viewpoint of GRAND and corresponding theoretical results.

- 2. In Table 1, we have replaced the GRAND++ results with a slightly fine-tuned $T$.

- 3. We have added more results on the ogbn-arxiv benchmark task, which requires using deep architectures. See Appendix D.6 in the revision.

- 4. We have performed the t-test to show the statistical significance of the accuracy gain of GRAND++ over GRAND. See Appendix D.5 in the revision.

- 5. We have added more discussion of Laplace learning and Poisson learning and contrasted them with GRAND and GRAND++ in the related work section.

-----

We are glad to answer any further questions you have on our submission.

---

### Author Response · Authors · 2021-11-19
**Any questions before the end of the discussion period?**

We would like to thank again all reviewers for their thoughtful reviews and valuable feedback.

We would appreciate it if you could let us know if there are additional questions or concerns before the end of the discussion period.

We would be happy to do any follow-up discussion or address any additional comments.

---

### Decision · Program_Chairs · 2022-01-20

**Decision:**

Accept (Poster)

**Comment:**

The paper presents a continuous framework for GNNs based on neural diffusion PDE and is an evolution of a previous method (GRAND). The main novelty appears to be the additional source term, which the author show to be beneficial in reducing the oversmoothing effect typical in deep GNNs. While novelty is somewhat limited, the paper provided detailed theoretical and experimental assessment of the idea. Overall, the reviewers liked the approach and expressed some questions/concerns that were satisfactorily addressed in the rebuttal. We recommend acceptance.